# Adaptive Rank, Reduced Forgetting: Knowledge Retention in Continual Learning Vision-Language Models with Dynamic Rank-Selective LoRA

## Abstract

Continual learning (CL) aims to accumulate knowledge from sequential tasks without catastrophic forgetting. Vision–language models like CLIP, with strong generalization, are widely used for CL. Existing methods often adapt isolated PTM components, adding inference complexity and limiting PTM improvement, or rely on replay, stored information, or assumptions, incurring high costs and limited applicability. To advance models as continual learners, we explore CL via natural, efficient PTM updates instead of complex task-specific additions. We thus study continual low-rank learning and systematically analyze how LoRA ranks and placements affect learning and forgetting. We find that a relatively higher-rank LoRA improves task learning (*i.e.*, *plasticity*) but increases forgetting, while a relatively lower-rank LoRA reduces forgetting (*i.e.*, *stability*) but limits adaptation. Crucially, we find a *plasticity–stability balance* tied to rank across parameters and tasks, with *moderately small ranks* maximizing CL benefits. Motivated by this, we propose **C**ontinual **Dy**namic **R**ank-Selective LoR**A** (**CoDyRA**), which continually updates PTMs with LoRA adapters of adaptively optimized rank. While the new-task objective drives learning, CoDyRA adaptively minimizes ranks with sparsity-promoting regularization to reduce interference and forgetting, achieving a plasticity–stability balance tailored to different parameters and tasks. Adaptively selected and minimized LoRA ranks keep the updated model closer to its previous state while learning new tasks. CoDyRA enables efficient CL as a sequence of LoRA-based tasks without storing past data, task information, or relying on assumptions. It preserves the original model architecture and deployment pipeline, adding no inference overhead. Extensive experiments show CoDyRA improves new representations while retaining old knowledge, achieving state-of-the-art results.

## 1 Introduction

Continual learning (CL) (Hadsell et al., 2020; De Lange et al., 2021; Wang et al., 2024b) focuses on incrementally learning from new data streams without catastrophic forgetting (Nguyen et al., 2019; McCloskey & Cohen, 1989) or the need to retrain on all previously seen data. With the rise of large-scale pre-trained models (PTMs), recent studies increasingly perform CL in this context (Zhou et al., 2024; McDonnell et al., 2024; Wang et al., 2024a; Jha et al., 2024). Many PTM-based CL approaches primarily focus on continual adaptation for downstream tasks (*e.g.*, class-incremental learning, CIL (Smith et al., 2023; Wortsman et al., 2022)), with an emphasis on downstream tasks. While task-specific and downstream-oriented designs (Zhou et al., 2024; McDonnell et al., 2024; Xu et al., 2024; Yu et al., 2024) show promising results on particular CL tasks, PTMs remain largely isolated from the CL process, mitigating forgetting but undermining interaction with new tasks and thus the potential for continual improvement. To advance toward models that function as continual learners, we consider CL with PTMs that enable flexible and efficient updates, preserving or even enhancing generalizable representations while acquiring new knowledge.

Leveraging the strong generalization and flexibility of jointly learned vision–text embeddings, CLIP as a vision–language model has been widely adopted in CL (Jha et al., 2024; Zheng et al., 2023; Wang

et al., 2023a; Zhou et al., 2023), including continual updating (Garg et al., 2023) and multi-domain CL (Zheng et al., 2023). Beyond downstream performance, the retention of knowledge in PTMs is also evaluated. Some approaches, such as ZSCL (Zheng et al., 2023), aim to continually update PTM, *i.e.*, CLIP, through full fine-tuning while mitigating forgetting via memory replay (Li & Hoiem, 2017; Wortsman et al., 2022; Rebuffi et al., 2017b) using additional reference data. However, this leads to high training costs and still struggles with catastrophic forgetting (Fig. 1(a)). To directly control the forgetting, some methods (Yu et al., 2024; Xu et al., 2024) introduce task-specific components isolated from the frozen PTMs, requiring task/domain prediction or gating at inference. These methods can deliver strong results with task-specific gating and branching; however, by isolating tasks and relying heavily on per-task modules and accurate task prediction, they become restricted to certain domains and limited in both general CL applicability and inference efficiency. Moreover, with restricted interaction between PTMs and CL process (*e.g.*, isolation) limits PTMs' generalization improvement potential (Fig. 1(b)).

Straightforward adaptation of PTMs can be achieved through direct model updating through full fine-tuning (FT) or parameter-efficient fine-tuning (PEFT) methods (Zhou et al., 2022c;b), such as Low-Rank Adaptation (LoRA) (Hu et al., 2021). This raises a fundamental question: instead of relying on complex additions or task-specific components, *can CL be performed by leveraging these natural PTM update mechanisms*? Our work investigates how to mitigate catastrophic forgetting within these standard frameworks without imposing extensive assumptions. We propose an efficient CL approach for CLIP that enables the model to be updated naturally across a stream of tasks, while retaining foundational pre-trained knowledge. We build upon LoRA for efficiency. While there have been some LoRA-based CL methods (Liang & Li, 2024; Wu et al., 2025; Zhu et al., 2025), they rely on specific regularization (*e.g.*, orthogonal property) for specific settings.

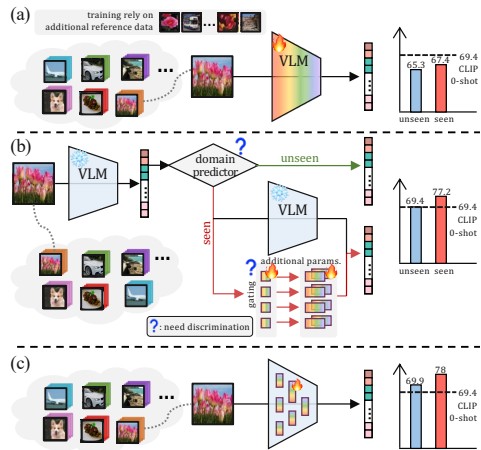

Figure 1: (a) Continual full fine-tuning leverages reference data to reduce forgetting of pre-trained knowledge (*e.g.*, zero-shot performance (Zheng et al., 2023)) but remains costly and prone to forgetting. (b) Task-specific modular methods (*e.g.*, (Yu et al., 2024; Xu et al., 2024)) add isolated components requiring domain prediction or gating, increasing complexity and limiting adaptability and generalization. (c) Our approach enables efficient, universal continual PTM updating without reference data or task-specific designs, retaining PTM knowledge and improving performance on unseen data.

We first systematically analyze and ablate how LoRA adapters at different locations in the network with different ranks influence learning and forgetting (Sec. 3.2). Our analyses reveal that *not all* LoRA ranks and placements contribute *equally* to downstream learning and forgetting. Our analyses empirically demonstrate that, for arbitrary weights, relatively higher-rank LoRA facilitates learning new tasks (*i.e.*, *plasticity*) but tends to increase forgetting, while lower-rank LoRA mitigates forgetting (*i.e.*, *stability*) but limits adaptation. Importantly, we identify that there exists a ***balance*** between ***plasticity*** (**learning new knowledge**) and ***stability*** (**regularizing forgetting**) at a moderately small (but not too small) rank, potentially maximizing CL benefits. Moreover, we observe that this balance *varies* significantly across different parameter placements.

The analysis results motivate us to propose **Co**ntinual **Dy**namic **R**ank-Selective LoRA (**CoDyRA**), which continually updates the PTM using LoRA adapters with dynamically and adaptively optimized ranks. CoDyRA performs new task learning while jointly minimizing the rank of each LoRA adaptively. Rank selection is guided by sparsity-promoting regularization and learnable importance weights, which dynamically adjust the contribution of each rank during training. By updating with LoRA adapters of minimized ranks, CoDyRA inherently biases models toward remaining closer to their previous state, thereby alleviating forgetting in principle without task-specific assumptions.

Our contributions are fourfold: (1) We study PTM-based CL with natural and efficient continual update of the model, not relying on complex task-specific additions, without storing past data, task information, or relying on assumptions. (2) We comprehensively analyze learning–forgetting trade-

offs across LoRA ranks and placements, identifying a plasticity–stability balance at moderately small ranks and deriving design strategies. This study guides our CL approach with regularized forgetting and provides evidence for future research. (3) We propose CoDyRA, which adaptively minimizes LoRA ranks via sparsity-promoting regularization. It handles different parameters and tasks, yielding regularized updates that reduce interference and keep the model closer to its previous state with less forgetting. (4) Extensive experiments across multiple CL benchmarks validate our method's superior performance in both downstream learning and mitigating forgetting, achieving a balanced trade-off between plasticity and stability.

## 2 RELATED WORK

**Continual Learning.** Continual learning enables models to sequentially learn new tasks while retaining previously acquired knowledge. Experience replay (ER) methods (Luo et al., 2023; Aljundi et al., 2019b; Chaudhry et al., 2018a; Liu et al., 2020; Chaudhry et al., 2018b; Yan et al., 2022; 2021) store subsets of past data to refresh the model on prior tasks during new training. Parameter regularization approaches (Kirkpatrick et al., 2017; Aljundi et al., 2018; Zenke et al., 2017; Aljundi et al., 2019a; Jha et al., 2023) constrain updates to important weights to preserve past knowledge. Meanwhile, dynamic networks (Wang et al., 2022a;b; Zhou et al., 2022a; Wang et al., 2024a; McDonnell et al., 2024; Liang & Li, 2024; Wang et al., 2022e;d; Smith et al., 2023; Wang et al., 2022c) adjust their architecture on the fly, balancing the acquisition of new information with the retention of old knowledge. TreeLoRA (Qian et al., 2025) routes inputs to task-specific adapters via a gradient-similarity tree and bandit search.

**Continual Learning of CLIP.** Continual learning of CLIP (Jha et al., 2024; Zhang et al., 2024b) aims to enable PTMs to sequentially learn across diverse domains while retaining their pre-trained generalization capabilities for previously seen tasks. (Garg et al., 2023) continually adapt CLIP to temporal distribution shifts via timestamped data updates. The approach introduced in ZSCL (Zheng et al., 2023) addresses the challenge of preserving zero-shot capabilities while adapting to new tasks, mitigating the risk of catastrophic forgetting. Subsequent works (Yu et al., 2024; Tang et al., 2025; Xu et al., 2024; Huang et al., 2025) have focused primarily on continual adaptation to downstream tasks while leveraging pre-trained predictions for unseen data. Recently, (Xu et al., 2024) proposed a more challenging setting, where the labels from different domains are mixed during testing.

**Low-Rank Adaptation.** Low-Rank Adaptation (LoRA) (Hu et al., 2021) has gained popularity as a parameter-efficient method for fine-tuning large pre-trained models. Similar to LoRA, (Wu et al., 2024b) learns a small set of task-specific representation vectors at each layer. Building on LoRA's framework, recent works enhance LoRA by reformulating parameter updates, focusing on initialization based on the pre-trained weights (Zhang et al., 2024a; Meng et al., 2024), treating each rank independently (Ding et al., 2023; Zhang et al., 2023; Liu et al., 2024), or employing a mixture of subspaces (Wu et al., 2024a). These advancements highlight LoRA's effectiveness in efficiently fine-tuning large models while preserving generalization.

## 3 METHODOLOGY

### 3.1 PRELIMINARIES

**Transformers.** A typical transformer model is composed of stacked blocks, each containing two main submodules: an multi-head attention (MHA) module (Attn) and a multilayer perceptron (MLP). The attention module at the $l$-th layer is defined as: $\text{Attn}(x) = \text{MHA}(x\mathbf{W}_l^{\text{Q}}, x\mathbf{W}_l^{\text{K}}, x\mathbf{W}_l^{\text{V}})\mathbf{W}_l^{\text{O}}$. The input sequence is denoted as $x \in \mathbb{R}^{n \times d}$, where $n$ is the sequence length and $d$ is the hidden dimension. The matrices $\mathbf{W}_l^{\text{Q}}, \mathbf{W}_l^{\text{K}}, \mathbf{W}_l^{\text{V}}, \mathbf{W}_l^{\text{O}} \in \mathbb{R}^{d \times d}$ correspond to the Query, Key, Value, and Output projection matrices, respectively. The MLP in each transformer block consists of two linear layers separated by an activation function $\text{Act}(\cdot)$, such as ReLU or GELU (Hendrycks & Gimpel, 2016). The MLP module at the $l$-th transformer layer is defined as: $\text{MLP}(x) = \text{Act}(x\mathbf{W}_l^{\text{FC}} + b_l^{\text{FC}})\mathbf{W}_l^{\text{Proj}} + b_l^{\text{Proj}}$, where $\mathbf{W}_l^{\text{FC}} \in \mathbb{R}^{d \times d_m}$ and $\mathbf{W}_l^{\text{Proj}} \in \mathbb{R}^{d_m \times d}$ are the weight matrices, with $d_m$ being the hidden dimension.

**CLIP model.** In CLIP (Radford et al., 2021; Ilharco et al., 2021; Fang et al., 2024; Li et al., 2022), both the image encoder $f_\theta$ and text encoder $g_\psi$ are transformers that learn a joint vision–language

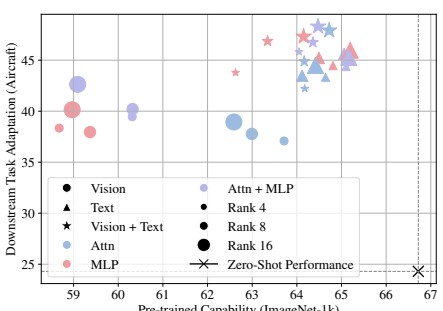

Figure 2: Overview of CoDyRA: we propose dynamic rank-selection LoRA, enabling each pre-trained weight matrix to adaptively add necessary ranks for downstream adaptation while retaining pre-trained capabilities. After each task, dynamic rank updates are merged into the pre-trained weights with no inference overhead.

embedding. Given an image $\mathbf{x} \in \mathbb{R}^{H \times W \times C}$, the vision encoder produces features $\mathbf{z}_V = f_\theta(\mathbf{x}) \in \mathbb{R}^D$. A class label $y$ is embedded into a prompt such as "a photo of a [CLS]" to form text input $\mathbf{t}$, which the text encoder maps to $\mathbf{z}_T = g_\psi(\mathbf{t}) \in \mathbb{R}^D$. At inference, the probability of classifying $\mathbf{x}$ into class $y_i \in \{1, \ldots, C\}$ is: $p(y_i|\mathbf{x}) = \frac{\exp(\mathtt{sim}(\mathbf{z}_V, \mathbf{z}_T^{y_i})/\tau)}{\sum_{c=1}^{C} \exp(\mathtt{sim}(\mathbf{z}_V, \mathbf{z}_T^{y_c})/\tau)}$, where $\mathtt{sim}(\cdot)$ is cosine similarity and $\tau$ is a temperature parameter.

**CL with CLIP.** We study continual learning of CLIP with multi-domain data, *e.g.*, multi-domain task-incremental learning (MTIL) (Zheng et al., 2023; Yu et al., 2024) and cross-domain task-agnostic incremental learning (X-TAIL) (Xu et al., 2024). In MTIL, the model sequentially learns from $T$ tasks, where each task $t$ has dataset $\mathcal{D}^t = \{(\mathbf{x}_i^t, y_i^t)\}_{i=1}^{N^t}$, with input image $\mathbf{x}_i^t \in \mathbb{R}^{H \times W \times C}$, label $y_i^t \in \mathcal{C}^t$, and $M^t$ classes in $\mathcal{C}^t = \{y_j^t\}_{j=1}^{M^t}$. In X-TAIL, the test-time category set includes both seen and unseen domains, $\mathcal{C} = \mathcal{C}_{\text{seen}} \cup \mathcal{C}_{\text{unseen}}$, where $\mathcal{C}_{\text{seen}} = \bigcup_{i=1}^{n} \mathcal{C}_i$ contains all classes from prior tasks and $\mathcal{C}_{\text{unseen}}$ denotes novel classes never encountered.

### 3.2 ANALYSES ON LoRA LOCATION AND RANK

To enable efficient and natural continual updating of PTMs, we employ LoRA and analyze its learning and forgetting capability for optimizing its usage. We examine how LoRA's rank and placement affect downstream learning and forgetting through two types of analyses (Pearl, 2022; Meng et al., 2022), providing a comprehensive view of their overall impact. **(1)** We first apply LoRA components with varying ranks at different locations in the pre-trained CLIP, for measurement of importance or contribution (Fig. 3); **(2)** We apply LoRA across all PTM weights and systematically prune them for evaluation, for testing the standalone capability (Fig. 4). We train the model on downstream datasets and assess its performance on both **(1)** the new task, *i.e.*, learning new knowledge, and **(2)** an additional reference dataset to evaluate the retention and forgetting of existing capabilities. We achieve three main results to guide methodology design. Without loss of generality, the learning–forgetting analyses can be extended to each learning step in straightforward, assumption-free CL.

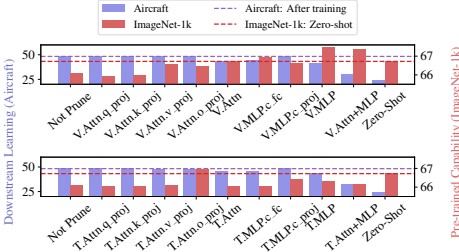

Figure 3: Task adaptation and zero-shot capability retention after training CLIP with different LoRA insertion points and ranks. Shapes indicate encoders, colors denote transformer modules, and sizes reflect rank values.

Figure 4: Impact of removing trained LoRA modules on downstream task adaptation (Aircraft, left y-axis) and retention of existing capabilities (ImageNet-1k, right y-axis). "T" and "V" indicate text-encoder and vision-encoder modules, respectively.

**Applying LoRA on all weights.** Unlike prior works (Hu et al., 2021; Zhang et al., 2024a; Meng et al., 2024) that only apply LoRA to limited components by default, we first examine the impact

of different LoRA placements. As shown in Fig. 3 and Fig. 4, applying LoRA to only specific modules alone may limit learning capacity in different ways: **(1)** LoRA placement strongly impacts downstream performance and retention of pre-trained capabilities; **(2)** Updating only the vision encoder causes more forgetting; **(3)** Updating only attention modules mitigates forgetting but limits learning on new tasks. These observations reflect the distinct roles of PTM components. **Takeaway 1:** Instead of manual or fixed LoRA placement (Hu et al., 2021; Zhang et al., 2024a; Meng et al., 2024), we apply LoRA to all weights and optimize for an *adaptive* configuration.

**The effects of LoRA ranks on balancing learning and forgetting.** We identified the rank number of LoRA plays a critical role for balancing learning and forgetting. As shown in Fig. 3, we can observe that a relatively higher-rank LoRA facilitates learning new tasks but tends to increase forgetting, while a lower-rank LoRA mitigates forgetting but limits adaptation. This suggests that lower-rank LoRA interferes less with existing knowledge (*i.e.*, reduces forgetting) while learning less new knowledge. Fortunately, we identify a *balance* (top-right in Fig. 3), where a moderately small rank reduces forgetting while enabling enough adaptability.

During training, low-rank parameter updates tend to enable the updated model to be closer to the original model, with less interference and forgetting. In the extreme case, a rank-zero LoRA learns nothing and introduces no forgetting. Thus, minimizing LoRA rank for each new task in CL can reduce forgetting without extra assumptions. On the other hand, the task-learning objective emphasizes new knowledge acquisition, enabling a balance. **Takeaway 2:** A plasticity–stability **balance** exists and associates with LoRA rank (high vs. low), which can be adaptively achieved by jointly optimizing the task objective and minimizing LoRA ranks.

**The balance varies by location and task.** Fig. 3 shows that the optimal rank for balance depends on both parameter placement and the learning task, consistent with the pruning-based sufficiency analysis in Fig. 4. For example, removing updates to the Value projection in the Attention module of the vision encoder had little impact on adaptation but substantially restored pre-trained performance. Interestingly, pruning the MLP layer slightly reduced task performance while enhancing pre-trained capabilities. **Takeaway 3:** The learning–forgetting balance point tied to rank varies across modules and tasks, necessitating adaptive optimization.

### 3.3 Proposed CoDyRA with Sparsity-Induced Rank Selection and Minimization

**Overview.** Our analyses in Sec. 3.2 suggest applying LoRA across all CLIP modules while adaptively optimizing for an appropriate rank for each specific task and location to balance learning and forgetting. Together with the task learning objective, minimizing active LoRA ranks acts as a regularizer, keeping the updated model closer to its previous state and alleviating forgetting. However, the balanced rank between learning and forgetting is not consistent across parameters or tasks (**Takeaway 3**). We address this with a sparsity-promoting regularization for the LoRA adapters.

We introduce adaptive rank-selective updates for all pre-trained weight matrices in the Attention and MLP modules of both vision and text encoders (Fig. 2). The sparsity-promoting regularization enables adaptive rank selection and minimization for each LoRA adapter on each task. For each task in CL, beyond adaptive rank minimization, CoDyRA requires no additional priors or assumptions and does not store or maintain information (*e.g.*, data, parameters, or statistics) from previous steps, unlike prior methods (Zheng et al., 2023; Yu et al., 2024; Xu et al., 2024).

During training on task $t$, we denote the pre-trained weights, or those updated from previous tasks, as $\{\mathbf{W}_0^{t,m}\}_{m=1}^M$, where $M$ is the total number of weight matrices updated by LoRA. After each task, the dynamic rank updates are merged back into the original weights.

**Regularized Low-rank updates with rank importance.** To update weight matrix $\mathbf{W}_0^{t,m} \in \mathbb{R}^{d \times k}$, LoRA (Hu et al., 2021) introduces two low-rank matrices, $\mathbf{B}^{t,m} \in \mathbb{R}^{d \times r}$ and $\mathbf{A}^{t,m} \in \mathbb{R}^{r \times k}$, such that $\Delta \mathbf{W}^{t,m} = \mathbf{B}^{t,m} \mathbf{A}^{t,m}$, where $r \ll \min(d, k)$. The updated weight matrix is then defined as:

$$\mathbf{W}^{t,m} = \mathbf{W}_0^{t,m} + \Delta \mathbf{W}^{t,m} = \mathbf{W}_0^{t,m} + \mathbf{B}^{t,m} \mathbf{A}^{t,m}. \tag{1}$$

Only $\mathbf{B}^{t,m}$ and $\mathbf{A}^{t,m}$ are trained for each task and then merged into the weights. LoRA incurs no additional computational overhead during inference.

As shown in Sec. 3.2, minimizing the active rank for each LoRA adapter alleviates forgetting, which can be formulated as minimizing a regularizer $R(\Delta \mathbf{W}^{t,m}) \doteq R(\mathbf{B}^{t,m} \mathbf{A}^{t,m})$. The minimum of

$R(\Delta\mathbf{W}^{t,m})$ is zero, *i.e.*, $\min R(\Delta\mathbf{W}^{t,m}) = 0$, corresponding to a rank-zero matrix—no update to $\mathbf{W}^{t,m}$ and hence no forgetting. By jointly optimizing the new-task learning objective and the minimized $R(\Delta\mathbf{W}^{t,m})$, we achieve a balance between learning and forgetting.

To automatically and adaptively minimize the active ranks in LoRA, we formulate $R(\Delta\mathbf{W}^{t,m})$ through a rank-selection LoRA. We introduce a learnable importance weight vector $\mathbf{w}^{t,m} \in \mathbb{R}^r$ to indicate the active ranks for a LoRA adapter:

$$\Delta\mathbf{W}^{t,m} = \sum\nolimits_{i=1}^{r} \mathbf{w}_i^{t,m}\mathbf{B}_{:,i}^{t,m}\mathbf{A}_{i,:}^{t,m} \tag{2}$$

where $\mathbf{w}_i^{t,m}$ is the learnable importance weight associated with rank $i$, $\mathbf{B}_{:,i}^{t,m}$ denotes the $i$-th column of $\mathbf{B}^{t,m}$, and $\mathbf{A}_{i,:}^{t,m}$ denotes the $i$-th row of $\mathbf{A}^{t,m}$. Minimizing $R(\Delta\mathbf{W}^{t,m})$ can be achieved through optimizing $\mathbf{w}^{t,m}$.

**Adaptive rank-selective updates with sparse-prompting regularization.** The formulation in Eq. (2) enables the model to dynamically optimize the importance of each rank and minimize active ranks through gradient descent.[1] To only retain the essential ranks for learning new tasks (thus reducing forgetting), we can achieve optimizing $R(\Delta\mathbf{W}^{t,m})$ through an $\ell_1$ norm-based sparsity-promoting regularization on $\mathbf{w}^{t,m}$, *i.e.*, $\|\mathbf{w}^{t,m}\|_1$. The optimization objective for parameters $\{\mathbf{w}^{t,m}, \mathbf{B}^{t,m}, \mathbf{A}^{t,m}\}_{m=1}^{M}$ learned on task $t$ is:

$$\mathcal{L}_{\text{train}}^t := \mathcal{L}_{\text{sup}}^t + \lambda\sum\nolimits_{m=1}^{M}\|\mathbf{w}^{t,m}\|_1, \tag{3}$$

where $\mathcal{L}_{\text{sup}}^t$ is the supervised training loss for current task, $M$ is the number of weight matrices inserted $r$-rank update, and $\mathbf{w}^{t,m}$ represents the importance weights of each rank added to the weight matrix $\mathbf{W}_0^{t,m}$. The $\ell_1$ regularization strength is controlled by $\lambda$.

To handle the non-differentiable $\ell_1$ regularization applied to the importance weights, we adopt the proximal gradient method (Beck & Teboulle, 2009). The $i$-th element of $\hat{\mathbf{w}}^{t,m}$ for rank $i$ is updated via soft-thresholding:

$$\mathbf{w}_i^{t,m} := \mathbb{1}(|\hat{\mathbf{w}}_i^{t,m}| > \kappa) \cdot (\hat{\mathbf{w}}_i^{t,m} - \text{sign}(\hat{\mathbf{w}}_i^{t,m}) \cdot \kappa), \tag{4}$$

where $\hat{\mathbf{w}}_i^{t,m}$ denotes the value of $\mathbf{w}_i^{t,m}$ after applying the gradient update from the supervised loss $\mathcal{L}_{\text{sup}}^t$. The threshold $\kappa$ increases from zero to $\kappa_{\max}$, analogous to linearly scheduling $\lambda$, and adaptively induces sparsity in $\mathbf{w}$ based on the task learning objective. The indicator $\mathbb{1}(\cdot)$ returns 1 if the condition holds and 0 otherwise, while $\text{sign}(\cdot)$ gives the input's sign ($\pm$). More details are left in Appendix A.

This approach preserves only the ranks with significant importance for learning current task and prunes those with low relevance, adapting across weights and tasks. The importance weights on preserved ranks softly highlight their relative significance. Early in training, dense updates are applied without Eq. (4), allowing all ranks to capture task-relevant information before sparsity is enforced.

**Continually learn on new tasks.** CoDyRA views CL as a sequence of tasks with LoRA, without storing past data Zheng et al. (2023), task information Yu et al. (2024); Xu et al. (2024), or relying on assumptions (Liang & Li, 2024). For each new task, new CoDyRA modules are initialized and optimized; low-importance ranks are pruned, and non-zero ranks merged into the weights. Sparsity-promoting regularization controls task capacity, keeping the model close to its previous state and reducing forgetting. This enables efficient continual updating with only sparsity-based regularization.

## 4 EXPERIMENTS

### 4.1 EXPERIMENTAL SETTING

**Datasets.** We evaluate our method on two widely used continual learning settings for vision-language models: MTIL (Zheng et al., 2023; Yu et al., 2024) and X-TAIL (Xu et al., 2024). A wide range of fine-grained datasets is used, where each dataset is treated as an individual task to be incrementally learned. We leave detailed experimental settings in Appendix B.

---

[1]To align the training process with the pre-trained model, we use CLIP's contrastive loss as the optimization objective during pre-training.

Table 1: Comparisons on X-TAIL for each domain. The **best** and the **second best** results are highlighted in **red** and **blue**, respectively. Methods marked with † indicate the use of domain prediction or distribution detection (Yu et al., 2024; Xu et al., 2024).

| Method | Aircraft | Caltech101 | DTD | EuroSAT | Flowers | Food | MNIST | OxfordPet | Cars | SUN397 | *Average* |
|---|---|---|---|---|---|---|---|---|---|---|---|
| *CLIP* | | | | | | | | | | | |
| Zero-shot | 23.5 | 76.8 | 37.3 | 36.7 | 63.6 | 84.0 | 46.7 | 86.7 | 66.1 | 63.7 | 58.5 |
| *Transfer* | | | | | | | | | | | |
| Zero-shot (Radford et al., 2021) | – | **76.8** | **37.3** | 36.7 | **63.6** | 84.0 | **46.7** | 86.7 | **66.1** | **63.7** | **62.4** |
| LwF (Li & Hoiem, 2017) | – | 66.6 | 26.9 | 19.5 | 51.0 | 78.4 | 26.6 | 68.9 | 35.5 | 56.1 | 47.7 |
| WiSE-FT (Wortsman et al., 2022) | – | 70.1 | 31.9 | 25.3 | 56.3 | 79.8 | 29.9 | 74.9 | 45.6 | 56.8 | 52.3 |
| iCaRL (Rebuffi et al., 2017b) | – | 71.7 | 35.0 | **43.0** | 63.4 | **86.9** | 43.9 | **87.8** | 63.7 | 60.0 | 61.7 |
| ZSCL (Zheng et al., 2023) | – | 73.3 | 32.6 | 36.8 | 62.1 | 83.8 | 42.1 | 83.6 | 56.5 | 60.2 | 59.0 |
| MoE-Adapter† (Yu et al., 2024) | – | 71.0 | 34.9 | 19.2 | 63.0 | **86.6** | 20.0 | 87.2 | 63.7 | 58.6 | 56.0 |
| RAIL-Primal† (Xu et al., 2024) | – | **76.8** | **37.3** | 36.7 | **63.6** | 84.0 | **46.7** | 86.7 | **66.1** | **63.7** | **62.4** |
| CoDyRA | – | 74.3 | 36.8 | **44.2** | **69.9** | 83.5 | 42.8 | **88.9** | **64.6** | 63.4 | **63.2** |
| CoDyRA† | – | 74.3 | 36.8 | **44.2** | **69.9** | 83.5 | 42.8 | **88.9** | **64.6** | 63.4 | **63.2** |
| *Average* | | | | | | | | | | | |
| LwF (Li & Hoiem, 2017) | 24.7 | 79.7 | 38.3 | 36.9 | 63.9 | 81.0 | 36.5 | 71.9 | 42.7 | 56.7 | 53.2 |
| WiSE-FT (Wortsman et al., 2022) | 27.1 | 76.5 | 40.9 | 31.3 | 68.7 | 81.6 | 31.4 | 74.7 | 51.7 | 58.4 | 54.2 |
| iCaRL (Rebuffi et al., 2017b) | 25.4 | 72.1 | 37.5 | 51.6 | 65.1 | **87.1** | 59.1 | 88.0 | 63.7 | 60.1 | 61.0 |
| ZSCL (Zheng et al., 2023) | 36.0 | 75.0 | 40.7 | 40.5 | 71.0 | 85.3 | 46.3 | 83.3 | 60.7 | 61.5 | 60.0 |
| MoE-Adapter† (Yu et al., 2024) | **43.6** | 77.9 | 52.1 | 34.7 | 75.9 | **86.3** | 45.2 | 87.4 | 66.6 | 60.2 | 63.0 |
| RAIL-Primal† (Xu et al., 2024) | 42.4 | **89.8** | 55.7 | 68.5 | **84.0** | 83.3 | **65.3** | 85.8 | **67.9** | **64.5** | 70.7 |
| CoDyRA | 41.4 | 81.0 | **58.7** | **77.8** | **83.4** | 84.6 | 64.5 | **90.4** | 67.2 | 64.4 | **71.3** |
| CoDyRA† | **43.9** | 81.6 | **60.6** | **78.4** | **84.0** | 84.9 | **64.6** | **90.5** | 67.4 | 64.4 | **72.0** |
| *Last* | | | | | | | | | | | |
| LwF (Li & Hoiem, 2017) | 25.5 | 72.1 | 38.9 | 55.4 | 65.5 | **87.3** | 81.9 | 88.6 | 63.6 | 61.5 | 64.0 |
| WiSE-FT (Wortsman et al., 2022) | 21.8 | 76.8 | 42.9 | 20.8 | 77.5 | 84.9 | 30.7 | 76.6 | 75.8 | 72.5 | 58.0 |
| iCaRL (Rebuffi et al., 2017b) | 25.5 | 72.1 | 38.9 | 55.4 | 65.5 | **87.3** | 81.9 | 88.6 | 63.6 | 61.5 | 64.0 |
| ZSCL (Zheng et al., 2023) | 33.1 | 75.3 | 43.5 | 35.2 | 74.6 | **87.4** | 50.4 | 84.2 | 77.3 | 73.4 | 63.4 |
| MoE-Adapter† (Yu et al., 2024) | **43.2** | 78.7 | 57.6 | 32.8 | 79.4 | 86.0 | 86.7 | 87.8 | **78.2** | **74.2** | 70.5 |
| RAIL-Primal† (Xu et al., 2024) | 41.7 | **94.0** | **66.0** | 86.4 | **97.2** | 82.4 | 93.1 | 83.6 | 75.0 | 71.3 | 79.1 |
| CoDyRA | 37.7 | 81.5 | 65.1 | **89.9** | 91.4 | 85.5 | **96.8** | **93.3** | 77.3 | **73.5** | **79.2** |
| CoDyRA† | **43.9** | 82.4 | **66.6** | **93.0** | 93.3 | 86.3 | **97.2** | **94.0** | **78.5** | **73.5** | **80.9** |

Table 2: Average transfer accuracy on unseen datasets not encountered during CL.

| Method | Average Transfer Accuracy on Unseen Data | | | |
|---|---|---|---|---|
| | CIFAR100 | Places365 | ImageNet-1k | *Average* |
| Zero-Shot (Radford et al., 2021) | 68.24 | 33.77 | 66.72 | 56.24 |
| MoE-Adapter (Yu et al., 2024) | 68.24 | 33.77 | 66.72 | 56.24 |
| RAIL (Xu et al., 2024) | 68.24 | 33.77 | 66.72 | 56.24 |
| **CoDyRA** | **68.95** | **36.52** | **68.52** | **58.00** |

**Evaluation Metrics.** We use the same metrics as prior work (Zheng et al., 2023; Yu et al., 2024; Xu et al., 2024): "Transfer" for zero-shot performance on unseen data, "Last" for retention of earlier tasks, and "Average" for mean accuracy across all datasets of all learning tasks.

**Implementation Details.** We follow the setups in (Zheng et al., 2023; Yu et al., 2024; Xu et al., 2024) using CLIP with a ViT-B/16 backbone (Radford et al., 2021). By default, CoDyRA is applied to all pre-trained weight matrices in the vision and text encoders with an initial rank of 16. Each task is trained for 500 iterations with AdamW (Loshchilov & Hutter, 2017), using dense training ratios of 0.5 for X-TAIL and 0.7 for MTIL. $\kappa_{\max}$ is set as 0.005.

## 4.2 EXPERIMENTAL RESULTS

**Cross-domain task-agnostic incremental learning.** We report X-TAIL results in Table 1. Unlike prior methods (Yu et al., 2024; Xu et al., 2024) that rely on zero-shot predictions to preserve pre-trained capabilities, our approach continually updates the model with minimal forgetting. As those methods are bounded by zero-shot performance, ours surpasses this limit, achieving higher transfer accuracy. While continual updates may slightly affect some unseen predictions, the overall effect is positive: the model adapts to new tasks while modestly improving generalization.

For the Last metric, our method achieves superior or comparable performance without domain ID prediction (Yu et al., 2024), high-dimensional projections, or memory banks (Xu et al., 2024). These techniques (Yu et al., 2024; Xu et al., 2024) are orthogonal and can be integrated into our

Table 3: Comparisons on 5-shot MTIL setting.

| Method | Aircraft | Caltech101 | CIFAR100 | DTD | EuroSAT | Flowers | Food | MNIST | OxfordPet | Cars | SUN397 | *Average* |
|---|---|---|---|---|---|---|---|---|---|---|---|---|
| *CLIP* | | | | | | | | | | | | |
| Zero-shot (Radford et al., 2021) | 24.3 | 88.4 | 68.2 | 44.6 | 54.9 | 71.0 | 88.5 | 59.4 | 89.0 | 64.7 | 65.2 | 65.3 |
| *Transfer* | | | | | | | | | | | | |
| Zero-shot (Radford et al., 2021) | – | 88.4 | 68.2 | 44.6 | 54.9 | 71.0 | 88.5 | 59.6 | 89.0 | 64.7 | 65.2 | 69.4 |
| LwF (Li & Hoiem, 2017) | – | 72.1 | 49.2 | 35.9 | 44.5 | 41.1 | 66.6 | 50.5 | 69.0 | 19.0 | 51.7 | 50.0 |
| LwF-VR (Ding et al., 2022) | – | 82.2 | 62.5 | 40.1 | 40.1 | 56.3 | 80.0 | 60.9 | 77.6 | 40.5 | 60.8 | 60.1 |
| WiSE-FT (Wortsman et al., 2022) | – | 77.6 | 60.0 | 41.3 | 39.4 | 53.0 | 76.6 | 58.1 | 75.5 | 37.3 | 58.2 | 57.7 |
| ZSCL (Zheng et al., 2023) | – | 84.0 | 68.1 | 44.8 | 46.8 | 63.6 | 84.9 | 61.4 | 81.4 | 55.5 | 62.2 | 65.3 |
| MoE-Adapter† (Yu et al., 2024) | – | 87.9 | 68.2 | 44.1 | 48.1 | 64.7 | 88.8 | 69.0 | 89.1 | 64.5 | 65.1 | 68.9 |
| RAIL-Primal† (Xu et al., 2024) | – | 88.4 | 68.2 | 44.6 | 54.9 | 71.0 | 88.5 | 59.6 | 89.0 | 64.7 | 65.2 | 69.4 |
| CoDyRA | – | 92.4 | 68.4 | 45.8 | 54.5 | 69.6 | 87.4 | 65.2 | 88.5 | 64.2 | 64.5 | 69.9 |
| CoDyRA† | – | 92.4 | 68.4 | 45.8 | 54.5 | 69.6 | 87.4 | 65.2 | 88.5 | 64.2 | 64.5 | 69.9 |
| *Average* | | | | | | | | | | | | |
| LwF (Li & Hoiem, 2017) | 23.5 | 77.4 | 43.5 | 41.7 | 43.5 | 52.2 | 54.6 | 63.4 | 68.0 | 21.3 | 52.6 | 49.2 |
| LwF-VR (Ding et al., 2022) | 24.9 | 89.1 | 64.2 | 53.4 | 54.3 | 70.8 | 79.2 | 66.5 | 79.2 | 44.1 | 61.6 | 62.5 |
| WiSE-FT (Wortsman et al., 2022) | 32.0 | 87.7 | 61.0 | 55.8 | 68.1 | 69.3 | 76.8 | 71.5 | 77.6 | 42.0 | 59.3 | 63.7 |
| ZSCL (Zheng et al., 2023) | 28.2 | 88.6 | 66.5 | 53.5 | 56.3 | 73.4 | 83.1 | 56.4 | 82.4 | 57.5 | 62.9 | 64.4 |
| MoE-Adapter† (Yu et al., 2024) | 30.0 | 89.6 | 73.9 | 58.7 | 69.3 | 79.3 | 88.1 | 76.5 | 89.1 | 65.3 | 65.8 | 71.4 |
| RAIL-Primal† (Xu et al., 2024) | 32.9 | 94.5 | 69.9 | 58.1 | 71.8 | 84.4 | 88.5 | 70.4 | 89.0 | 66.1 | 65.7 | 71.9 |
| CoDyRA | 34.6 | 95.8 | 73.9 | 60.0 | 77.1 | 81.3 | 86.6 | 75.9 | 89.9 | 66.1 | 65.3 | 73.3 |
| CoDyRA† | 37.6 | 96.0 | 76.6 | 62.1 | 78.7 | 82.0 | 86.8 | 76.0 | 90.0 | 66.2 | 65.3 | 74.3 |
| *Last* | | | | | | | | | | | | |
| LwF (Li & Hoiem, 2017) | 22.1 | 58.2 | 17.9 | 32.1 | 28.1 | 66.7 | 46.0 | 84.3 | 64.1 | 31.5 | 60.1 | 46.5 |
| LwF-VR (Ding et al., 2022) | 22.9 | 89.8 | 59.3 | 57.1 | 57.6 | 79.2 | 78.3 | 77.7 | 83.6 | 60.1 | 69.8 | 66.9 |
| WiSE-FT (Wortsman et al., 2022) | 30.8 | 88.9 | 59.6 | 60.3 | 80.9 | 81.7 | 77.1 | 94.9 | 83.2 | 62.8 | 70.0 | 71.9 |
| ZSCL (Zheng et al., 2023) | 26.8 | 88.5 | 63.7 | 55.7 | 60.2 | 82.1 | 82.6 | 58.6 | 85.9 | 66.7 | 70.4 | 67.4 |
| MoE-Adapter† (Yu et al., 2024) | 30.1 | 89.3 | 74.9 | 64.0 | 82.3 | 89.4 | 87.1 | 89.0 | 89.1 | 69.5 | 72.5 | 76.1 |
| RAIL-Primal† (Xu et al., 2024) | 32.9 | 95.1 | 70.3 | 63.2 | 81.5 | 95.6 | 88.5 | 89.7 | 89.0 | 72.5 | 71.0 | 77.2 |
| CoDyRA | 31.6 | 95.5 | 72.8 | 63.5 | 85.0 | 89.7 | 85.0 | 94.7 | 93.2 | 73.6 | 73.0 | 78.0 |
| CoDyRA† | 37.6 | 96.4 | 78.4 | 68.2 | 92.6 | 92.3 | 86.2 | 94.9 | 93.8 | 75.2 | 73.0 | 80.8 |

Table 4: Comparison with a broad range of CL methods on the TRACE benchmark using the LLaMA-3.2-1B-Instruct. We report Overall Performance (OP (%) ↑) and Backward Transfer (BWT (%) ↓).

| | FIX(ICL) | SeqLoRA | OGD | GEM | EWC | L2P | DualPrompt | HiDeLoRA | O-LoRA | TreeLoRA | CoDyRA |
|---|---|---|---|---|---|---|---|---|---|---|---|
| OP (%) ↑ | 31.16 | 29.73 | 30.12 | 32.19 | 31.96 | 29.38 | 30.76 | 33.73 | 32.94 | 36.14 | **37.46** |
| BWT (%) ↓ | - | 17.03 | 15.2 | 10.74 | 11.62 | 13.57 | 11.34 | 12.36 | 12.89 | 7.36 | **5.11** |

framework, aligning with our goal of leveraging natural PTM updates. In practice, we use domain-wise autoencoders to predict test domains for CoDyRA†.

**Multi-domain task-incremental learning.** We further evaluate our method in the few-shot MTIL setting (Table 3), following (Yu et al., 2024; Xu et al., 2024). As with X-TAIL, our approach outperforms prior methods across all metrics without extra techniques and even surpasses their zero-shot upper bound in average transfer accuracy. Incorporating techniques from (Yu et al., 2024; Xu et al., 2024) further boosts performance.

**Experiments of continual learning on LLMs.** To evaluate the performance of our method on Large Language Models (LLMs), we employ the TRACE (Wang et al., 2023b) dataset, a benchmark specifically designed for studying continual learning in LLMs. Specifically, we evaluated LLaMA-3.2-1B-Instruct (Dubey et al., 2024) on the TRACE benchmark, reporting Overall Performance (OP) and Backward Transfer (BWT). The results, presented in Table 4, demonstrate that CoDyRA scales effectively to LLM architectures and achieves performance competitive with prior works.

## 4.3 DISCUSSIONS

**Pre-trained knowledge retention and potential pre-trained model improvement.** To assess transfer beyond the zero-shot limits of prior works, we evaluated on unseen datasets—CIFAR100 (Krizhevsky et al., 2009), Places365 (Zhou et al., 2017), and ImageNet-1k (Deng et al., 2009) (Table 2). Unlike methods that rely solely on zero-shot predictions (Yu et al., 2024; Xu et al., 2024) and are constrained by CLIP's baseline, our approach integrates new knowledge into the pre-trained space while preserving existing capabilities. This yields superior zero-shot transfer accuracy, demonstrating that CoDyRA effectively expands general knowledge and improves overall representations.

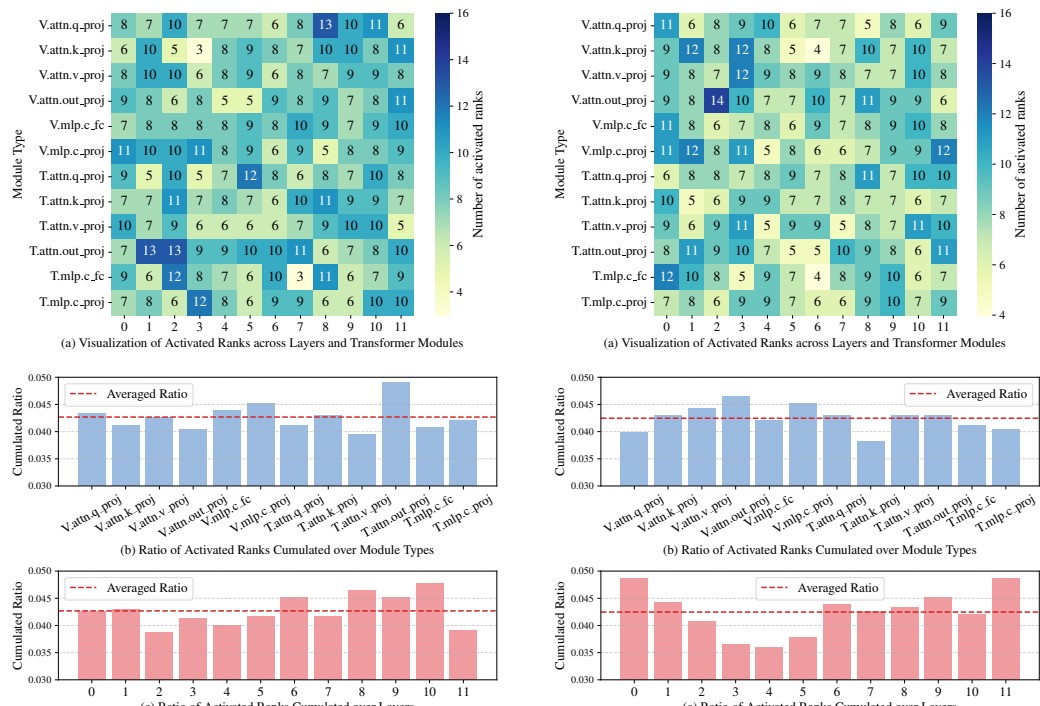

Figure 5: Visualization and statistical analysis of rank activation on the Aircraft dataset.

Figure 6: Visualization and statistical analysis of rank activation on the Oxford Pets dataset.

**Rank allocation across different datasets.** We visualize the ranks merged into the pre-trained weights for different datasets in Fig. 5 and Fig. 6 (more in Appendix Fig. 10 to Fig. 17.). The results show that CoDyRA adapts the number of active ranks differently across locations and tasks, reflecting the varying demands of each dataset. These strike a balance between plasticity and stability, demonstrating the method's ability to regulate forgetting while enabling effective task learning.

For the Aircraft dataset, we observe that a greater number of ranks are allocated to the Output projection of the Attention module within the text encoder (Fig. 5(b)). Additionally, a higher concentration of ranks is assigned to deeper layers, particularly layers 6 to 10 (Fig. 5(c)).

In contrast, for the Pets dataset, more ranks are allocated to the Attention module's Value and Output projections and the MLP's Projection layer within the vision encoder (Fig. 6(b)). Additionally, rank allocation is primarily concentrated in the first and last layers (Fig. 6(c)), indicating a distinct adaptation pattern that balances knowledge retention and adaptation across datasets. These observations highlight the adaptive nature of our approach, where rank allocation dynamically adjusts based on dataset characteristics, ensuring effective domain adaptation while preserving pre-trained knowledge. We provide more visualizations and analyses in Appendix B.4.

**Computation cost.** We compare computational costs in terms of trainable parameters and inference overhead, as shown in Table 5. For MoE-Adapter, the gap between training and testing parameters stems from its mixture-of-experts design: all experts are active during training, while only the top-2 are used at inference. RAIL-Primal introduces extra overhead from high-dimensional feature expansion via projection and regression layers. RAIL-Dual further increases

Table 5: Computation cost.

| Methods | Training Params. | Additional Infer. Params. / Mem. |
|---|---|---|
| LWF (Li & Hoiem, 2017) | 129.6M | None |
| ZSCL (Zheng et al., 2023) | 129.6M | None |
| MoE-Adapters (Yu et al., 2024) | 59.8M | 13.35M |
| RAIL (Xu et al., 2024) | N/A | 24.18M / 9.01M |
| **CoDyRA** | **4.4M** | **None** |

memory use by storing image features for all samples across 1,100 classes. These methods all incur linearly growing costs as tasks increase. In contrast, our method reduces training overhead by retaining only high-importance ranks; once merged into the pre-trained weights, the updates add zero memory cost at inference.

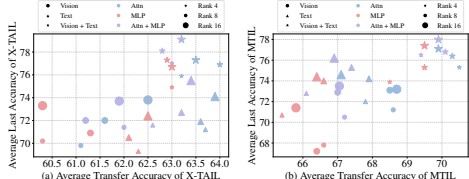 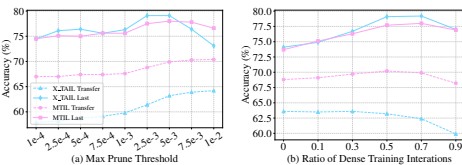

Figure 7: Ablations on different insertion locations and the impact of varying the initial rank.

Figure 8: Analyses (a) maximum pruning threshold and (b) ratios of dense training iterations.

## 4.4 ABLATION STUDIES

**Effects of insertion locations and initial ranks.** We study the impact of insertion locations and initial ranks (Fig. 7), consistent with our earlier analyses (Fig. 3). Training restricted to the vision encoder significantly reduces transfer accuracy, indicating degraded pre-trained capabilities. At certain locations, increasing rank improves adaptation but does not always hinder transfer accuracy.

Interestingly, restricting updates to only Attention modules can even surpass our reported state-of-the-art Transfer results (Table 1, Table 3), but at the cost of weaker adaptation. To balance adaptation and retention, we update across all modules by default, achieving an optimal trade-off between learning new tasks and preserving pre-trained knowledge.

**Analyses of pruning threshold and dense-iteration ratio.** To understand the trade-off between learning and forgetting, we vary the maximum threshold $\kappa_{max}$ and the dense training ratio (Fig. 8). Raising $\kappa_{max}$ improves Last accuracy by pruning noisy ranks, but overly aggressive pruning limits adaptation. For simplicity, we do not tune the value of $\kappa_{max}$ for each downstream task. The dense-iteration ratio controls pruning onset: pruning from the start (ratio = 0) removes ranks before they learn useful knowledge, while delaying too long risks retaining noisy ranks that do not get pruned.

## 5 CONCLUSION

We introduced CoDyRA, a continual learning framework for CLIP that dynamically selects and minimizes LoRA ranks based on their learned importance. Guided by sparsity-promoting regularization, CoDyRA balances plasticity with stability, without storing past data, task information, or relying on assumptions. By keeping updates close to the model's previous state while preserving essential ranks, our method mitigates forgetting and enhances representation quality. Extensive experiments across diverse continual learning benchmarks demonstrate that CoDyRA achieves state-of-the-art performance, effectively balancing downstream adaptation and knowledge retention.

**Future works.** CoDyRA updates the model continually by balancing learning and forgetting in the current task. While enabling a flexible and efficient CL process with strong results, future work will further refine modeling of rank relationships, inter-task dependencies, and forgetting measures.

REPRODUCIBILITY STATEMENT

Detailed experiment settings and implementation details are provided in Sec. 4.1 and Appendix B. The source code will be made publicly available upon acceptance of the paper.

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

THE USE OF LARGE LANGUAGE MODELS (LLMS)

Large language models (LLMs) were employed in preparing this manuscript, specifically to assist with grammar checking and polishing. All technical content, experimental design, and analysis were conceived and carried out by the authors.

# A ADDITIONAL DETAILS OF THE PROPOSED METHOD

In Sec. 3.3 of the main paper, we introduce our method for learning dynamic rank-selective parameter updates to pre-trained weight matrices. We leverage a learnable vector to represent the importance of each rank, which is updated through a soft-thresholding operation. In this section, we provide more details to enhance clarity and understanding.

## A.1 FULL DERIVATION OF EQ. (4)

For simplicity, we omit the notation for the training task $t$ in the following derivations. With a slight abuse of notation, we use subscript $t$ to represent $t$-th training iteration. At the $t$-th training iteration, the training loss with $\ell_1$ sparse regularization, as defined in the main paper, is:

$$\mathcal{L}_{\text{train}}(\Delta_t) := \mathcal{L}_{\text{sup}}(\Delta_t) + \lambda \sum_{m=1}^{M} \|\mathbf{w}_t^m\|_1, \tag{3}$$

where $\mathcal{L}_{\text{sup}}$ is the supervised training loss, $\Delta_t$ is the all trainable parameters $\{\mathbf{w}_t^m, \mathbf{B}_t^m, \mathbf{A}_t^m\}_{m=1}^{M}$ at $t$-th iteration, $M$ is the total number of pre-trained weight matrices with LoRA of rank $r$, $\mathbf{w}_t^m$ represents the importance weights of the $m$-th LoRA module, and $\lambda > 0$ controls the strength of the $\ell_1$ regularization.

We apply proximal gradient descent (Beck & Teboulle, 2009) to handle the non-differentiable $\ell_1$ regularization of the importance weights (Ding et al., 2023). At the $t$-th iteration, the update rule for the importance weights of each LoRA becomes:

$$\mathbf{w}_{t+1}^m \leftarrow \arg\min_{\mathbf{w}^m} \eta_t \cdot \lambda \|\mathbf{w}^m\|_1$$
$$+ \frac{1}{2}\|\mathbf{w}^m - (\mathbf{w}_t^m - \eta_t \nabla_{\mathbf{w}^m}\mathcal{L}_{\text{sup}}(\Delta_t))\|_2^2, \tag{5}$$

where $\eta_t > 0$ is the step-size of $t$-th iteration, and $\nabla_{\mathbf{w}^m}\mathcal{L}_{\text{sup}}(\Delta_t)$ is the gradient of the supervised loss with respect to $\mathbf{w}^m$. This update balances the $\ell_1$ sparsity regularization term $\|\mathbf{w}^m\|_1$, with the quadratic penalty term $\frac{1}{2}\|\mathbf{w}^m - \hat{\mathbf{w}}_t^m\|_2^2$, which ensures proximity to the gradient update if importance weights $\hat{\mathbf{w}}_t^m := \mathbf{w}_t^m - \eta_t \nabla_{\mathbf{w}^m}\mathcal{L}_{\text{sup}}(\Delta_t)$. This can be achieved through a soft-thresholding operator $\mathcal{T}$:

$$\mathbf{w}_{t+1}^m \leftarrow \mathcal{T}_{\eta_t \cdot \lambda}(\mathbf{w}_t^m - \eta_t \nabla_{\mathbf{w}^m}\mathcal{L}_{\text{sup}}(\Delta_t)), \tag{6}$$

where the soft-thresholding operator $\mathcal{T}$ is defined as:

$$\mathcal{T}_\kappa(x) = \mathbb{1}(|x| > \kappa) \cdot (x - \text{sign}(x) \cdot \kappa), \tag{7}$$

such that the input to the operator $\mathcal{T}_\kappa$ is the elements in $\hat{\mathbf{w}}_t^m$, $\hat{\mathbf{w}}_t^m$ is the value of $\mathbf{w}_t^m$ after the gradient update using only $\mathcal{L}_{\text{sup}}(\Delta_t)$, and $\kappa = \eta_t \cdot \lambda$ is the threshold. The indicator function $\mathbb{1}(\cdot)$ returns 1 if the condition is met, and 0 otherwise, while $\text{sign}(\cdot)$ denotes the sign $(+/-)$ of the input. This concludes our derivation of the dynamic update for the importance weights as presented in the main paper Eq. (4).

## A.2 MORE DISCUSSIONS ON ANALYTICAL RESULTS OF SEC. 3.2

### A.2.1 TAKEAWAY 1 - ON APPLYING LORA TO ALL MODULES

The original LoRA paper applies LoRA only to selected modules (mainly the attention components). Many subsequent LoRA and LoRA-based continual learning works (e.g., InfoLoRA, MoE-Adapters, O-LoRA, etc.) also apply LoRA to only part of the Transformer modules, though the specific choices vary. To clarify this design space, we conducted an analysis to determine which module configuration is most effective. Our results show that applying LoRA to all layers yields clear benefits.

While the theoretical analysis of standard Transformers and full LoRA training dynamics is complex, the empirical Neural Tangent Kernel (eNTK) (Malladi et al., 2023) offers a practical framework for theoretical investigation. The necessity of applying LoRA to all layers, rather than just attention modules, can be explained through the lens of the eNTK .

The eNTK of LoRA approximates that of Full Fine-Tuning (FullFT) only when LoRA is applied to the layers that contain the majority of the parameters. Since the kernel is based on the dot products of gradients ($K(i, j) = g_i \cdot g_j$), layers with more parameters (e.g., MLPs) typically exert the most influence on the kernel. Restricting updates to attention layers limits the model's ability to approximate the learning dynamics of FullFT, resulting in "slower learning" or reduced plasticity.

### A.2.2 TAKEAWAY 2 - ON THE RANK-DEPENDENT PLASTICITY-STABILITY BALANCE

We formalize the causality of this balance by defining the trade-off as an informational dilemma rooted in the subspace dimensionality:

**High Rank $\rightarrow$ High Plasticity / Low Stability.** The rank ($r$) dictates the intrinsic dimensionality (degrees of freedom) of the update subspace $\Delta W$. A high rank maximizes the informational capacity of the parameter shift. In the NTK context, maximizing rank pushes the update closer to the FullFT regime. While this allows for rapid acquisition of new task knowledge (High Plasticity), the larger informational perturbation maximizes disruption to the pre-trained knowledge manifold, leading to catastrophic forgetting (Low Stability).

**Low Rank $\rightarrow$ High Stability / Low Plasticity.** Conversely, restricting rank constrains the degrees of freedom. This forces the model to encode only the minimal, essential information for the new task. This minimization of total informational perturbation ensures the model remains informationally proximal to its previous state—a necessary condition for stability.

A lower rank $r$ of the LoRA implies a smaller dimension for the update $\Delta W$. A low rank acts as a structural regularizer. It forces the update to be the "simplest" change necessary to solve the current task. This minimizes the norm $||\Delta W||_F$ and reduces the probability that the update vector on the critical subspace of previous tasks.

### A.2.3 TAKEAWAY 3 - ON THE NECESSITY OF ADAPTIVE OPTIMIZATION

The need for adaptive rank selection stems from the functional variance of model components and the different information requirements of tasks.

Following the theory on intrinsic dimension of objective landscapes (Li et al., 2018), different tasks require different degrees of freedom to be solved. And, not all layers contribute equally to the objectives (eNTK or feature transformation in eNTK theory). Adaptivity allows the model to allocate higher ranks to layers with high gradient influence (e.g., specific MLPs) while enforcing sparsity on less critical layers.

### A.3 MORE DISCUSSIONS ON RELATED WORKS

We provide additional comparisons and clarifications of related work that complement the discussion in the main paper:

1. CLAP4CLIP (Jha et al., 2024). This framework uses probabilistic variational inference to model task-specific distributions, which introduces additional complexity from probabilistic modeling and sampling. CoDyRA enforces structural sparsity directly in the parameter space, yielding a more stable and computationally efficient optimization procedure.

2. C-CLIP (Liu et al., 2025). This method utilizes an auxiliary Contrastive Knowledge Consolidation (CKC) objective function to preserve zero-shot performance. Unlike C-CLIP, which relies on external loss constraints to enforce retention, CoDyRA relies on intrinsic parameter regulation ($l_1$ sparsity).

3. MG-CLIP (Mind the Gap) (Liang et al., 2022). This method adds auxiliary visual-space classifiers at inference time to mitigate the modality gap, altering the standard inference pipeline. CoDyRA preserves the original CLIP architecture and deployment, maintaining vision–language alignment by regulating the update rank, with no extra inference overhead or external modules.

4. CL-LoRA (He et al., 2025). This method adopts a dual-adapter design with separate task-shared and task-specific modules plus gradient reassignment, leading to higher architectural and parameter-management complexity. CoDyRA is significantly simpler, using a single unified adapter per layer and achieving the plasticity–stability trade off via dynamic rank selection (pruning non-essential ranks) rather than architectural bifurcation.

5. InfLoRA (Liang & Li, 2024). This method projects updates into a subspace strictly orthogonal to prior tasks to eliminate interference. This enforces a "hard" constraint that can overly restrict the solution space and limit plasticity. CoDyRA applies a "soft" constraint via sparsity-promoting regularization.

6. ConvPrompt (Roy et al., 2024).

   (a) Fundamental mechanism:
       i. ConvPrompt (Input-conditioned weights): Its weights are dynamically calculated at inference time by conditioning on the input features (e.g., using [CLS] embeddings to query prompt keys via cosine similarity). These weights change for every single image to perform instance-aware prompt combination.
       ii. CoDyRA (Input-Independent weights): Our importance weights are global learnable parameters optimized directly via gradient descent and soft thresholding. They are not conditioned on input features. Instead, they serve as a importance score for each rank of updates, determining the effective rank of the update matrix $\Delta W$.

   (b) Usage: Because ConvPrompt relies on input conditioning, it must compute these weights on-the-fly for every sample, introducing inference overhead. In contrast, CoDyRA's weights are structural parameters that are merged into the linear weights after training. This results in zero inference overhead, maintaining the original efficiency of the pre-trained backbone.

   (c) Functionality: ConvPrompt's weights act as an attention mechanism for input adaptation. CoDyRA's weights, guided by $l_1$ sparsity, act as a capacity control mechanism to balance plasticity and stability by pruning redundant ranks. Thus, CoDyRA represents a method for parameter-efficient fine-tuning, which is technically and functionally distinct from the conditional prompting mechanism in ConvPrompt.

## B  MORE EXPERIMENTAL RESULTS AND ABLATIONS

**Further details of the experiment settings.** The MTIL setting consists of 1,201 classes drawn from 11 diverse datasets: Aircraft (Maji et al., 2013), Caltech101 (Fei-Fei et al., 2004), CIFAR100 (Krizhevsky et al., 2009), DTD (Cimpoi et al., 2014), EuroSAT (Helber et al., 2019), Flowers (Nilsback & Zisserman, 2008), Food (Bossard et al., 2014), MNIST (Deng, 2012), Oxford-Pet (Parkhi et al., 2012), Cars (Krause et al., 2013), and SUN397 (Xiao et al., 2010). In the X-TAIL setting, a total of 10 datasets are used, with CIFAR100 (Fei-Fei et al., 2004) excluded to prevent domain overlap, following the protocol in (Xu et al., 2024). In line with (Xu et al., 2024), we use a 5-shot split for MTIL and a 16-shot split for X-TAIL. For initialization, we follow the standard convention: matrix $\mathbf{A}$ is randomly initialized, matrix $\mathbf{B}$ is set to zero, and the importance weights $\mathbf{w}$ are randomly initialized.

### B.1  MORE EXPERIMENT RESULTS

#### B.1.1  COMPARISON TO FIXED-RANK LORA BASELINES.

Table 6 reports results for fixed-rank LoRA (r = 4, 8, 16) and the LoRA variants, including LAE (Gao et al., 2023), TreeLoRA (Qian et al., 2025), InfLoRA (Liang & Li, 2024) and CL-LoRA (He et al., 2025). Several key observations emerge: **(1)** As LoRA rank increases, Last accuracy improves while Average accuracy declines, aligning with Fig. 3: a relatively higher-rank LoRA facilitates learning new tasks but tends to increase forgetting, while lower-rank LoRA mitigates forgetting but limits adaptation. **(2)** Fixed-rank LoRA shows a substantial Last-accuracy gap compared to our adaptive method. This suggests that without dynamic rank selection, low-relevance ranks (potentially encoding noise) are retained in the model, impairing generalization to unseen domains and accumulating inter-task interference over time, resulting in severe performance drop. **(3)** Leveraging fast online LoRA updates, slow EMA aggregation in offline LoRA, and energy-based logit ensembling, LAE

Table 6: Extended results for Table 1 on X-TAIL for each domain, comparing continual fine-tuning with different LoRA settings and other LoRA variants.

| Method | Aircraft | Caltech101 | DTD | EuroSAT | Flowers | Food | MNIST | OxfordPet | Cars | SUN397 | Average |
|---|---|---|---|---|---|---|---|---|---|---|---|
| *CLIP* | | | | | | | | | | | |
| Zero-shot | 23.5 | 76.8 | 37.3 | 36.7 | 63.6 | 84.0 | 46.7 | 86.7 | 66.1 | 63.7 | 58.5 |
| *Transfer* | | | | | | | | | | | |
| Zero-shot (Radford et al., 2021) | – | **76.8** | 37.3 | 36.7 | 63.6 | **84.0** | **46.7** | 86.7 | **66.1** | **63.7** | 62.4 |
| LoRA (r=4) | – | 74.9 | **39.4** | 41.8 | 67.7 | 83.3 | 44.1 | 87.1 | 64.2 | 62.7 | 62.8 |
| LoRA (r=8) | – | 77 | 38.2 | 38.4 | **67.8** | 83.3 | **44.6** | **87.6** | 64 | 62.8 | 62.6 |
| LoRA (r=16) | – | 76.4 | 39.1 | 37.5 | 65.9 | 82.7 | 41.9 | 86.9 | 63.4 | 62.7 | 61.8 |
| LAE-LoRA (Gao et al., 2023) | – | 76.2 | **39.9** | **43.8** | 66.5 | 83.3 | 42.7 | 87.5 | 63 | 62.9 | **62.9** |
| TreeLoRA (Qian et al., 2025) | - | 76.0 | 36.3 | 34.0 | 58.5 | 77.2 | 43.3 | 82.2 | 49.8 | 55.8 | 57.0 |
| InfLoRA (Liang & Li, 2024) | - | 75.8 | 34.5 | 29.2 | 58.1 | 73.4 | 38.6 | 79.5 | 47.7 | 50.3 | 54.1 |
| CL-LoRA (He et al., 2025) | - | 73.3 | 33.7 | 29.5 | 58.5 | 80.3 | 43.1 | 85.5 | 61.5 | 59.2 | 58.3 |
| CoDyRA | – | 74.3 | 36.8 | **44.2** | **69.9** | 83.5 | 42.8 | **88.9** | 64.6 | **63.4** | **63.2** |
| CoDyRA† | – | 74.3 | 36.8 | **44.2** | **69.9** | 83.5 | 42.8 | **88.9** | 64.6 | **63.4** | **63.2** |
| *Average* | | | | | | | | | | | |
| LoRA (r=4) | 23.5 | 78.4 | 40.6 | 47.3 | 71.4 | 83.7 | 50.1 | 87.5 | 65.1 | 64 | 61.2 |
| LoRA (r=8) | 23.7 | 78.1 | 39.7 | 46.6 | 71.3 | 83.7 | 50.7 | 87.8 | 65.4 | 64.1 | 61.1 |
| LoRA (r=16) | 23.6 | 78.5 | 40.5 | 43.0 | 70.4 | 82.8 | 49.5 | 87.5 | 64.9 | 64.0 | 60.5 |
| LAE-LoRA (Gao et al., 2023) | 29.2 | 81.3 | 50.7 | 68.8 | 77.4 | 83.8 | 53.2 | 88.5 | 66.3 | **64.2** | 66.4 |
| TreeLoRA (Qian et al., 2025) | 19.3 | 81.4 | 55.3 | 63.9 | 77.6 | 80.4 | 64.3 | 85.0 | 54.8 | 57.6 | 64.0 |
| InfLoRA (Liang & Li, 2024) | 38.3 | **84.2** | **60.1** | 45.8 | 79.0 | 77.5 | 61.8 | 82.8 | 52.2 | 52.2 | 63.4 |
| CL-LoRA (He et al., 2025) | 39.7 | 79.2 | 56.7 | 58.5 | 79.1 | 81.1 | 64.0 | 78.0 | 63.6 | 60.6 | 66.0 |
| CoDyRA | **41.4** | 81.0 | 58.7 | **77.8** | **83.4** | **84.6** | 64.5 | **90.4** | 67.2 | **64.4** | **71.3** |
| CoDyRA† | **43.9** | **81.6** | **60.6** | 78.4 | **84.0** | **84.9** | **64.6** | **90.5** | **67.4** | **64.4** | **72.0** |
| *Last* | | | | | | | | | | | |
| LoRA (r=4) | 9.75 | 79.5 | 40.7 | 37.5 | 65.1 | 82.6 | 47.1 | 84.1 | 57.5 | **76.3** | 58 |
| LoRA (r=8) | 12.7 | 78.7 | 39.6 | 42.6 | 62.4 | 82.3 | 46.8 | 83.6 | 60.2 | **76.1** | 58.5 |
| LoRA (r=16) | 13.7 | 78.6 | 40.3 | 39.4 | 66.1 | 82.7 | 50.9 | 85.7 | 60.3 | **76.1** | 59.4 |
| LAE-LoRA (Gao et al., 2023) | 25.6 | 81.5 | 50.2 | 73.2 | 79.5 | 85 | 70.4 | 91.1 | 72.9 | 75.7 | 70.5 |
| TreeLoRA (Qian et al., 2025) | 16.0 | 79.1 | 59.2 | 62.3 | 83.0 | 83.6 | 95.0 | 89.9 | 71.9 | 74.2 | 71.4 |
| InfLoRA (Liang & Li, 2024) | 38.3 | **85.2** | **66.6** | 52.9 | **92.9** | 81.6 | 96.6 | 90.4 | 70.1 | 69.3 | 74.4 |
| CL-LoRA (He et al., 2025) | **39.7** | 79.8 | 62.5 | 70.9 | **92.9** | 81.9 | 95.2 | 90.5 | 72.4 | 73.1 | 75.9 |
| CoDyRA | 37.7 | 81.5 | 65.1 | 89.9 | 91.4 | 85.5 | 96.8 | 93.3 | 77.3 | 73.5 | 79.2 |
| CoDyRA† | **43.9** | **82.4** | **66.6** | **93.0** | **93.3** | **86.3** | **97.2** | **94.0** | **78.5** | 73.5 | **80.9** |

substantially outperforms fixed-rank LoRA (r = 16). Nonetheless, unlike our method, fixed-rank LoRA lacks fine-grained control over the contribution of individual ranks within each module.

### B.1.2 ADDITIONAL RESULTS ON FULL-SHOT MTIL.

To further extend our evaluation, we report full-shot results on the MTIL benchmark (Table 7). For fairness, we adopt the same domain-prediction design used in recent methods. Consistent with Tables 1 and 3, CoDyRA achieves state-of-the-art performance, surpassing all baselines on both Transfer (unseen-domain generalization) and Last (continual downstream learning).

Table 7 also compares against prompt-based approaches (L2P (Wang et al., 2022e), Dual-Prompt (Wang et al., 2022d), S-Prompt (Wang et al., 2022c)). Although these methods mitigate task forgetting via key–value matching for prompt selection, they neglect the preservation of pre-trained capabilities—evidenced by substantial drops in transfer accuracy—and further underperform recent domain-prediction approaches on last accuracy.

### B.1.3 TRAINING STABILITY.

Table 8 reports the mean and standard deviation of CoDyRA's results in Table 1, averaged over three independent runs. The consistently low variance across metrics demonstrates that our method is stable to random initialization, highlighting CoDyRA's robust adaptive rank selection.

### B.1.4 ANALYSIS IN FORGETTING

We provide a dedicated forgetting analysis in Table 9, quantifying forgetting via Backward Transfer (BWT). It reports BWT of CoDyRA and some relevent CL methods with representation updating. Fixed-rank LoRA serves as a baseline and exhibits significant forgetting due to unconstrained updates without regularization. ZSCL relies on the replay of reference data to help mitigate forgetting.

Table 7: Comparisons on full-shot MTIL setting.

| Method | Aircraft | Caltech101 | CIFAR100 | DTD | EuroSAT | Flowers | Food | MNIST | OxfordPet | Cars | SUN397 | Average |
|---|---|---|---|---|---|---|---|---|---|---|---|---|
| *CLIP* | | | | | | | | | | | | |
| Zero-shot (Radford et al., 2021) | 24.3 | 88.4 | 68.2 | 44.6 | 54.9 | 71 | 88.5 | 59.4 | 89 | 64.7 | 65.2 | 65.2 |
| *Transfer* | | | | | | | | | | | | |
| Zero-shot (Radford et al., 2021) | – | 74.5 | 56.9 | 39.1 | **51.1** | 52.6 | 72.8 | 60.6 | 75.1 | 30.3 | 55.9 | 56.9 |
| LwF (Li & Hoiem, 2017) | – | 56.6 | 44.6 | 32.7 | 39.3 | 46.6 | 68.0 | 60.6 | 77.4 | 31.9 | 60.5 | 50.4 |
| iCaRL (Rebuffi et al., 2017a) | – | 77.1 | 61.0 | 40.5 | 45.3 | 54.4 | 74.6 | 47.9 | 76.7 | 36.3 | 58.6 | 57.2 |
| LwF-VR (Ding et al., 2022) | – | 73.5 | 55.6 | 35.6 | 41.5 | 47.0 | 68.3 | 53.9 | 69.3 | 26.8 | 51.9 | 52.3 |
| WiSE-FT (Wortsman et al., 2022) | – | 86.0 | 67.4 | **45.4** | 50.4 | 69.1 | **87.6** | 61.8 | 86.8 | 60.1 | **66.8** | 68.1 |
| ZSCL (Zheng et al., 2023) | – | 55.7 | 50.9 | 30.4 | 41.4 | 49.3 | 71.8 | 36.3 | 77.5 | 55.3 | 53.4 | 53.2 |
| L2P† (Wang et al., 2022e) | – | 55.7 | 50.9 | 30.4 | 41.4 | 49.3 | 71.8 | 36.3 | 77.5 | 55.3 | 53.4 | 53.2 |
| DualPrompt† (Wang et al., 2022d) | – | 66.7 | 51.4 | 28.7 | 33.7 | 45.6 | 70.9 | 59.5 | 77.7 | 49.5 | 50.4 | 52.4 |
| S-Prompts† (Wang et al., 2022c) | – | 67.3 | 49.4 | 26.4 | 39.7 | 47.1 | 70.2 | 34.3 | 78.9 | 56.7 | 52.2 | 52.2 |
| DIKI† (Tang et al., 2025) | – | **92.9** | **69.0** | 43.2 | 48.2 | 67.4 | 85.2 | **63.0** | 87.9 | 63.8 | **66.2** | 68.7 |
| MoE-Adapter† (Yu et al., 2024) | – | 87.9 | 68.2 | 44.4 | 49.9 | **70.7** | **88.7** | 59.7 | **89.1** | **64.5** | 65.5 | 68.9 |
| CoDyRA† | – | **92.4** | **68.8** | 45.2 | 50 | 69.4 | 84.2 | 62.3 | 88.8 | 64.6 | 65 | **69.1** |
| *Average* | | | | | | | | | | | | |
| LwF (Li & Hoiem, 2017) | 36.3 | 86.9 | 72.0 | 59.0 | 73.7 | 60.0 | 73.6 | 74.8 | 80.0 | 37.3 | 58.1 | 64.7 |
| iCaRL (Rebuffi et al., 2017a) | 35.5 | 89.2 | 72.2 | 60.6 | 68.8 | 70.0 | 78.2 | 62.3 | 81.8 | 41.2 | 62.5 | 65.7 |
| LwF-VR (Ding et al., 2022) | 29.6 | 87.7 | 74.4 | 59.5 | 72.4 | 63.6 | 77.0 | 66.7 | 81.2 | 43.7 | 60.7 | 65.1 |
| WiSE-FT (Wortsman et al., 2022) | 26.7 | 86.5 | 64.4 | 57.1 | 65.7 | 58.7 | 71.1 | 70.5 | 75.8 | 36.9 | 54.6 | 60.7 |
| ZSCL (Zheng et al., 2023) | 45.1 | 92.0 | 80.1 | 64.3 | 79.5 | 81.6 | **89.6** | 75.2 | 88.9 | 64.7 | **68.0** | 75.4 |
| L2P† (Wang et al., 2022e) | 38.0 | 85.2 | 78.2 | 61.3 | 72.9 | 74.9 | 79.7 | 59.1 | 82.0 | 59.7 | 55.4 | 67.9 |
| DualPrompt† (Wang et al., 2022d) | 37.8 | 84.3 | 78.5 | 60.1 | 71.1 | 73.2 | 79.1 | 73.9 | 82.3 | 55.1 | 52.8 | 68.0 |
| S-Prompts† (Wang et al., 2022c) | 37.5 | 92.5 | 77.5 | 58.2 | 76.4 | 74.1 | 78.8 | 57.9 | 83.0 | 60.8 | 54.4 | 68.3 |
| DIKI† (Tang et al., 2025) | 45.1 | **95.5** | **83.1** | 64.8 | 79.9 | **83.5** | 87.0 | **76.2** | **89.6** | 67.0 | **67.1** | 76.3 |
| MoE-Adapter† (Yu et al., 2024) | **50.2** | 91.9 | **83.1** | **69.4** | 78.9 | **84.0** | **89.1** | 73.7 | 89.3 | **67.7** | 66.9 | **76.7** |
| CoDyRA† | 48.3 | **96.4** | **83.2** | 69.1 | 80.0 | 84.0 | 86.0 | 75.6 | 90.5 | 67.5 | 66.3 | **77.0** |
| *Last* | | | | | | | | | | | | |
| LwF (Li & Hoiem, 2017) | 26.3 | 87.5 | 71.9 | 66.6 | 79.9 | 66.9 | 83.8 | **99.6** | 92.1 | 66.1 | 80.4 | 74.6 |
| iCaRL (Rebuffi et al., 2017a) | 35.8 | 93.0 | 77.0 | 70.2 | 83.3 | 88.5 | **90.4** | 86.7 | 93.2 | 81.2 | **81.9** | 80.1 |
| LwF-VR (Ding et al., 2022) | 20.5 | 89.8 | 72.3 | 67.6 | 85.5 | 73.8 | 85.7 | **99.6** | 93.1 | 73.3 | 80.9 | 76.6 |
| WiSE-FT (Wortsman et al., 2022) | 27.2 | 90.8 | 68.0 | 68.9 | 86.9 | 74.0 | 87.6 | **99.6** | 92.6 | 77.8 | **81.3** | 77.7 |
| ZSCL (Zheng et al., 2023) | 40.6 | 92.2 | 81.3 | 70.5 | 94.8 | 90.5 | **91.9** | 98.7 | 93.9 | **85.3** | 80.2 | 83.6 |
| L2P† (Wang et al., 2022e) | 38.0 | 87.1 | 84.2 | 72.9 | 86.0 | 96.1 | 89.2 | 99.0 | 94.1 | 79.6 | 76.0 | 82.0 |
| DualPrompt† (Wang et al., 2022d) | 37.8 | 87.1 | 84.6 | 71.8 | 89.2 | 96.3 | 89.1 | 99.1 | **94.5** | 79.9 | 76.5 | 82.3 |
| S-Prompts† (Wang et al., 2022c) | 37.5 | 95.1 | 83.7 | 70.2 | **97.5** | **96.5** | 89.0 | 99.1 | 94.0 | 79.5 | 75.8 | 83.4 |
| DIKI† (Tang et al., 2025) | 45.2 | **95.7** | 86.3 | 72.9 | **98.0** | **97.0** | 89.2 | **99.4** | 94.2 | **81.6** | 76.6 | **85.1** |
| MoE-Adapter† (Yu et al., 2024) | **49.8** | 92.2 | **86.1** | **78.1** | 95.7 | 94.3 | 89.5 | 98.1 | 89.9 | **81.6** | 80.0 | 85.0 |
| CoDyRA† | 48.3 | **96.8** | 85.9 | 77.8 | 97.5 | 96.2 | 88.3 | 99.0 | **94.9** | 80.6 | 78.6 | **85.8** |

Table 8: Statistical significance of CoDyRA results corresponding to Table 1.

| | Aircraft | Caltech101 | DTD | EuroSAT | Flowers | Food | MNIST | OxfordPet | Cars | SUN397 | Average |
|---|---|---|---|---|---|---|---|---|---|---|---|
| CoDyRA *Transfer* | – | $74.3^{\pm0.52}$ | $36.8^{\pm0.23}$ | $44.2^{\pm0.56}$ | $69.9^{\pm0.56}$ | $83.5^{\pm0.23}$ | $42.8^{\pm0.18}$ | $88.9^{\pm0.42}$ | $64.6^{\pm0.47}$ | $63.4^{\pm0.56}$ | $63.2^{\pm0.28}$ |
| CoDyRA *Average* | $41.4^{\pm0.28}$ | $81^{\pm0.38}$ | $58.7^{\pm0.26}$ | $77.8^{\pm0.47}$ | $83.4^{\pm0.39}$ | $84.6^{\pm0.28}$ | $64.5^{\pm0.14}$ | $90.4^{\pm0.4}$ | $67.2^{\pm0.23}$ | $64.4^{\pm0.47}$ | $71.3^{\pm0.18}$ |
| CoDyRA *Last* | $37.7^{\pm0.42}$ | $81.5^{\pm0.24}$ | $65.1^{\pm0.63}$ | $89.9^{\pm0.55}$ | $91.4^{\pm0.38}$ | $85.5^{\pm0.16}$ | $96.8^{\pm0.08}$ | $93.3^{\pm0.3}$ | $77.3^{\pm0.66}$ | $73.5^{\pm0.21}$ | $79.2^{\pm0.18}$ |

MoE-Adapters isolates task-wise knowledge within specific adapters and adopts domain prediction to further reduce interference. InfLoRA constrains parameter updates to a direction orthogonal to prior tasks to reduce interference. CoDyRA effectively mitigates forgetting by minimizing the rank of updates via sparsity regularization. This ensures the model learns effectively while remaining close to its original state, demonstrating strong retention capabilities with a BWT of only 1.87%.

Moreover, in Table 4, we validate our method in the language domain by continually training the LLaMA-3.2-1B-Instruct model. The low BWT scores observed in this setting further support our claims of reduced forgetting across architectures.

Table 9: Forgetting analysis. We report Backward Transfer (BWT) averaged across all X-TAIL tasks, where lower values indicate better retention.

| Methods | ZSCL | MoE-Adapter | InfLoRA | LoRA ($r$=4) | LoRA ($r$=8) | LoRA ($r$=16) | CoDyRA |
|---|---|---|---|---|---|---|---|
| BWT (%) ↓ | 8.78 | 5.43 | 3.12 | 13.94 | 14.24 | 14.86 | **1.87** |

Table 10: Experimental results for integrating aggression-based adapters in the X-TAIL setting. Metrics for "Transfer", "Average", and "Last" are reported as averages across all datasets.

|  | Transfer | Average | Last |
|---|---|---|---|
| RAIL-Primal | 62.4 | 70.7 | 79.1 |
| **CoDyRA** + RAIL-Primal | **63.1** | **74.1** | **83.4** |
| RAIL-Dual | 62.4 | 71.9 | 82.4 |
| **CoDyRA** + RAIL-Dual | **63.1** | **74.2** | **83.6** |

### B.2 EFFECTS OF ADDITIONAL ADAPTIVE PREDICTION HEAD ON VISUAL REPRESENTATIONS

The related continual learning methods focus on updating the representation with the backbone model (*e.g.*, (Wang et al., 2022e; Zheng et al., 2023; Yu et al., 2024), and CoDyRA) or newly added representation alignment or prediction head (McDonnell et al., 2024; Xu et al., 2024). These two types of methods can be seen as orthogonal. We study how the representation updated by the proposed can work with the additional adaptive prediction head, relying on the aggression-based adapters in (Xu et al., 2024).

We provide a more detailed analysis of the effects of the additional prediction head on the visual representations in X-TAIL setting in Table 10. Note that the Dual version of RAIL (Xu et al., 2024) stores all visual representations of all samples for all seen classes and domains in memory, it is not included in Tables 1 and 3 for a fair comparison.

RAIL is a regression-based method that applies an additional prediction head on top of the visual representations extracted by the frozen pre-trained vision encoder of the CLIP model. This approach is fully orthogonal to our method, as our focus is on continually incorporating new knowledge into the pre-trained model through dynamic low-rank parameter updates, and we can seamlessly integrate their techniques into our method. Because RAIL heavily relies on the frozen pre-trained model, the Transfer accuracy is limited by the zero-shot performance of the pre-trained model. In contrast, our approach gradually accumulates knowledge into the pre-trained model, enhancing its capability on unseen data and achieving improved Transfer accuracy. Moreover, by leveraging our continually enhanced pre-trained model, coupling our method with RAIL significantly outperforms the original RAIL, which uses a frozen pre-trained model, on the Average and Last metrics.

### B.3 ANALYSIS OF CHANGES/INTERFERENCE W.R.T. PRE-TRAINED WEIGHTS

In Sec. 4.3 of the main paper, Fig. 5 and Fig. 6 visualize and analyze the dynamic ranks assigned to each module. Here, we extend the analysis by comparing parameter changes to pre-trained weights using the amplification factor as in (Hu et al., 2021). The amplification factor is given by:

$$\text{Amp} = \frac{\|\Delta\mathbf{W}\|_F}{\|\mathbf{U}^{\mathrm{T}}\mathbf{W}\mathbf{V}^{\mathrm{T}}\|_F}, \tag{8}$$

where $\mathbf{W}$ is the pre-trained weight matrix, and $\Delta\mathbf{W}$ is the parameter change introduced by LoRA. For vanilla LoRA and CoDyRA, $\Delta\mathbf{W}$ is computed using Eq. (1) and Eq. (2), respectively. $\mathbf{U}$ and $\mathbf{V}$ represent the top $r$ singular vectors of $\text{SVD}(\Delta\mathbf{W})$, where $r$ is the number of interested ranks in analysis.

A higher amplification factor indicates that low-rank parameter updates amplify directions less emphasized in the pre-trained weights (Hu et al., 2021). We visualize the amplification factors for both our method and vanilla LoRA trained on the Aircraft, EuroSAT and Oxford Pets datasets in Fig. 9.

The results show that, compared to vanilla LoRA, our method tends to concentrate parameter updates on a more specific subset of transformer modules while reducing updates to pre-trained weights less relevant to the current data. This demonstrates that our method more effectively identifies critical pre-trained weight matrices relevant to the current data and applies targeted dynamic parameter updates. Additionally, by reducing the strength of parameter updates for pre-trained weights less related to the current data, our method minimizes the impact on pre-trained knowledge and knowledge acquired from previous tasks compared to vanilla LoRA.

Note that this analysis differs from the visualizations of number of activated ranks in Fig. 5 and Fig. 6. The number of activated ranks reflects the quantity of significant ranks contributing to parameter updates for leaning each task. In this analysis, the amplification factor directly measures the parameter change to each pre-trained weight matrix.

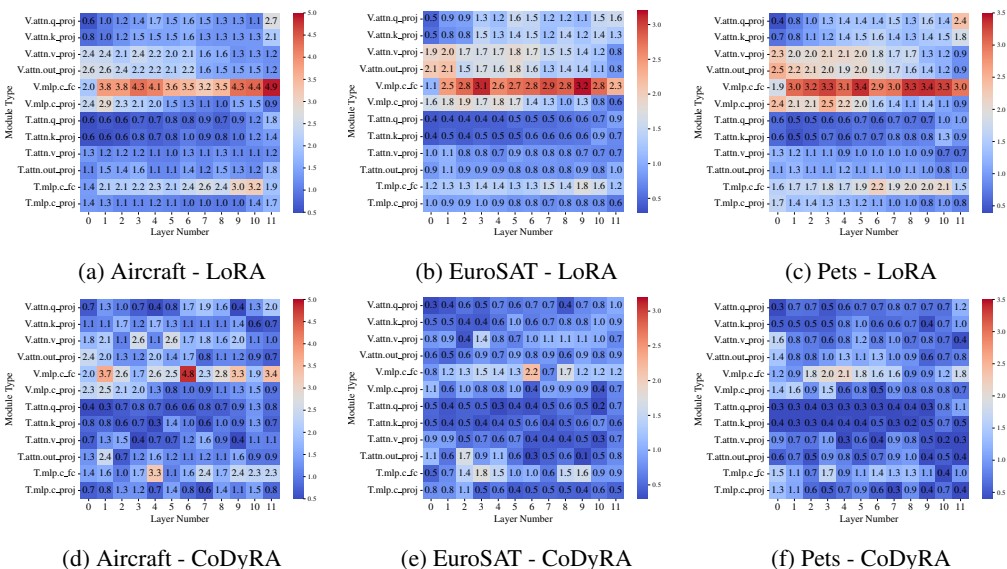

(a) Aircraft - LoRA  (b) EuroSAT - LoRA  (c) Pets - LoRA

(d) Aircraft - CoDyRA  (e) EuroSAT - CoDyRA  (f) Pets - CoDyRA

Figure 9: Visualization of amplification factors when trained on (a, d) Aircraft, (b, e) EuroSAT, and (c, f) Pets, comparing (d, e, f) our method of dynamic sparse rank selective LoRA and (a, b, c) vanilla LoRA. It shows that the proposed method with adaptive rank selection can achieve more focused/concentrated updating and less interference.

## B.4 MORE VISUALIZATIONS AND ANALYSES ON RANK ALLOCATION RESULTS OF DIFFERENT DATASETS

In Sec. 4.3, Fig. 5 and Fig. 6, we visualize and analyze the dynamic ranks assigned to each module for the Aircraft and Oxford Pets datasets under the X-TAIL experimental settings. Here, we extend these visualizations and analyses to additional datasets, as shown from Fig. 10 to Fig. 17.

The visualizations, along with statistical analyses of transformer modules and layers, reveal distinct rank allocation patterns across datasets. These findings suggest that the rank of parameter updates needed for achieving a good downstream improvements and knowledge retention is distinct across each kind of data.

## B.5 EVALUATION WITH OTHER VLMS

To evaluate the broader applicability of our method, we conducted preliminary experiments with BLIP (Li et al., 2022) on X-TAIL (Table 11). Our method consistently outperformed prior works and exceeded the pre-trained model's upper-bound performance, demonstrating its general effectiveness across different VLMs.

Table 11: Averaged performance using BLIP.

|  | Transfer | Average | Last |
|---|---|---|---|
| Zero-shot | 51.03 | – | 46.45 |
| RAIL-Primal† | 51.03 | 62.21 | 75.72 |
| **CoDyRA†** | **51.59** | **63.60** | **76.36** |

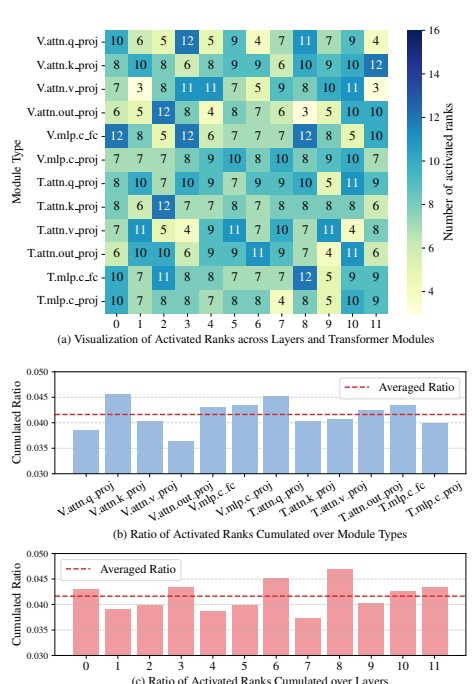

Figure 10: Visualization and statistical analysis of rank activation on the Caltech101 dataset using our proposed method.

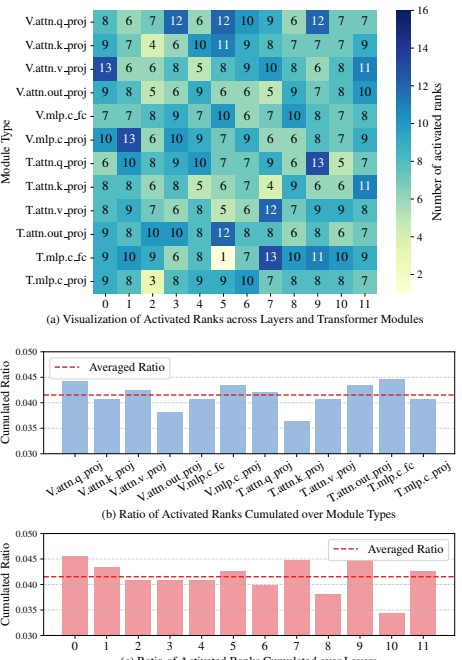

Figure 11: Visualization and statistical analysis of rank activation on the DTD dataset using our proposed method.

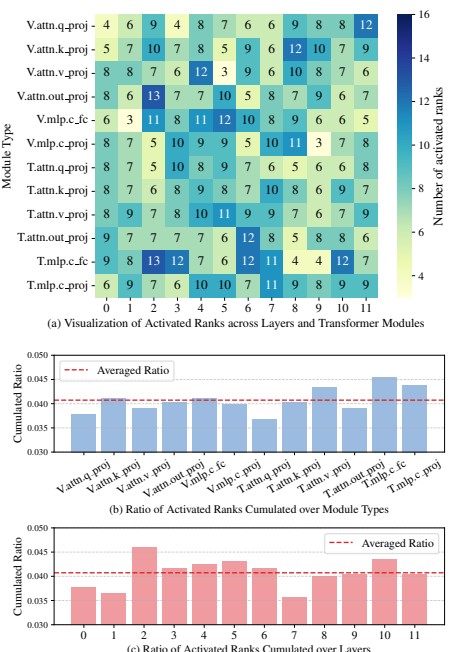

Figure 12: Visualization and statistical analysis of rank activation on the EuroSAT dataset using our proposed method.

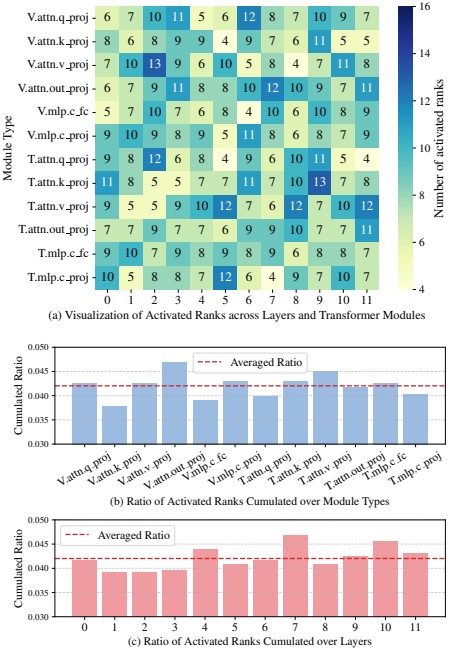

Figure 13: Visualization and statistical analysis of rank activation on the Flowers dataset using our proposed method.

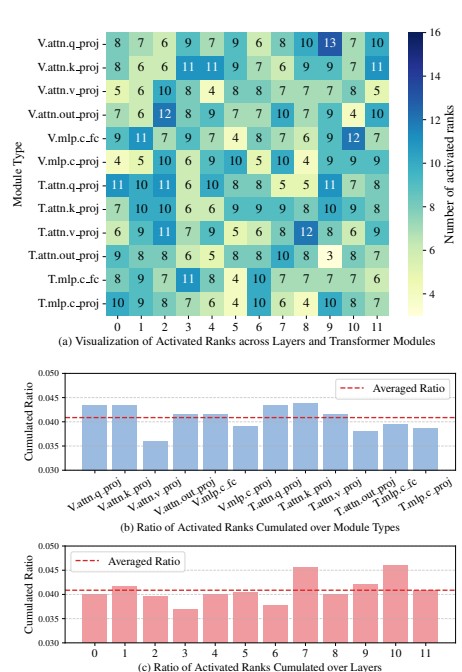

Figure 14: Visualization and statistical analysis of rank activation on the Food101 dataset using our proposed method.

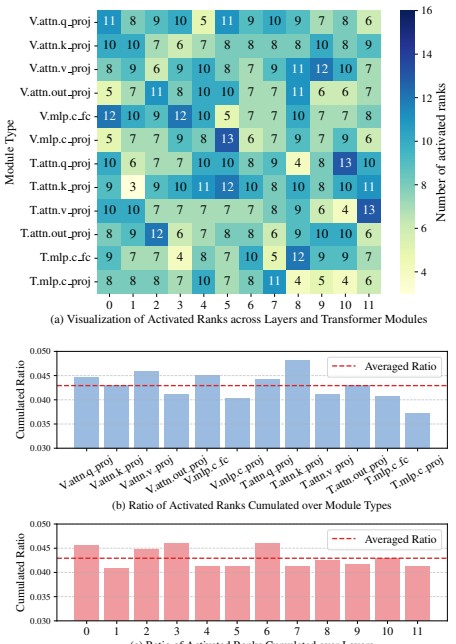

Figure 15: Visualization and statistical analysis of rank activation on the MNIST dataset using our proposed method.

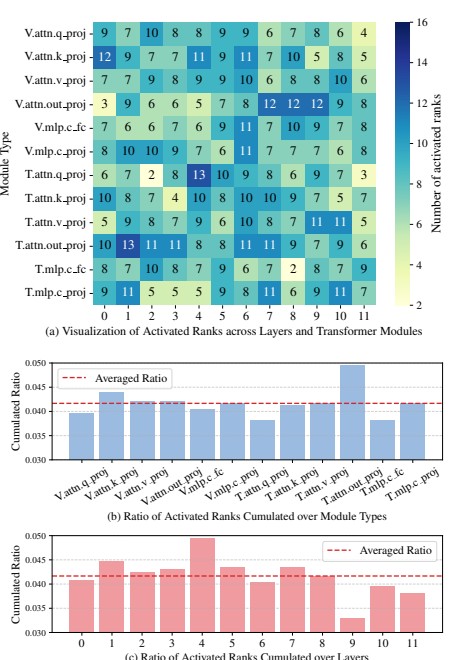

Figure 16: Visualization and statistical analysis of rank activation on the Stanford Cars dataset using our proposed method.

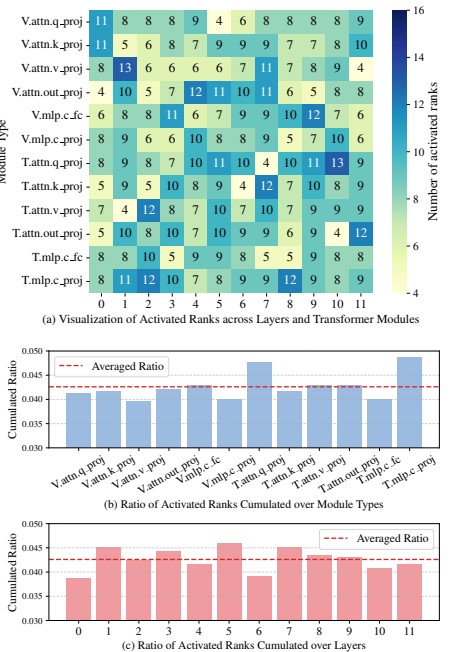

Figure 17: Visualization and statistical analysis of rank activation on the SUN397 dataset using our proposed method.

Table 12: Performance of 16-shot PEFT on downstream tasks.

|  | Acc. ↑ / Final Used Params. ↓ | | |
|---|---|---|---|
|  | Caltech101 | Cars | OxfordPet |
| Zero-shot | 88.40 | 64.70 | 89.00 |
| LoRA ($r = 16$) | 96.43 / 4.4M | 80.66 / 4.4M | 92.94 / 4.4M |
| **CoDyRA** ($r_{init} = 16$) | **97.20 / 3.95M** | **81.61 / 3.75M** | **94.44 / 3.85M** |

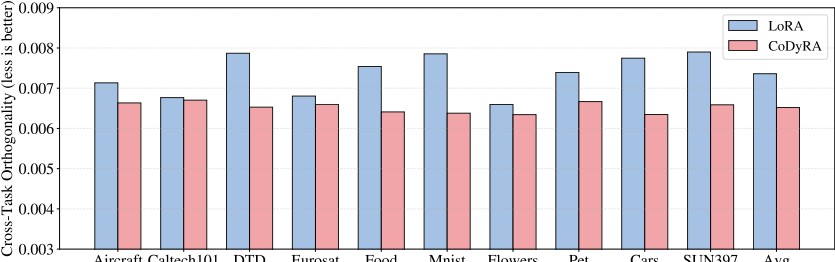

Figure 18: Parameter-space directional analysis of **A**-matrix updates. For each task, the column displays the average cosine similarity between the model's LoRA updates after a specific task and the LoRA updates from models trained on all other tasks. This analysis quantifies directional interference among sequential updates. Lower cosine similarity reflects more orthogonal (less interfering) updates. CoDyRA produces lower similarity than the simple LoRA–merging baseline, demonstrating stronger implicit orthogonality of parameter updates even without explicit orthogonal regularization.

## B.6    Evaluation on Standard Parameter-Efficient Fine-Tuning (PEFT) Task

Leveraging our dynamic rank-adaptive parameter updates, our method naturally extends to Parameter-Efficient Fine-Tuning (PEFT) tasks. Table 12 presents 16-shot PEFT results, using the same hyperparameters as the main paper, showing that CoDyRA outperforms vanilla LoRA while reducing final trainable parameters by pruning ranks with zero importance. Notably, across multiple downstream tasks (Caltech101, Cars, and OxfordPet), CoDyRA achieves higher accuracy while using fewer parameters compared to LoRA with a fixed rank of 16.

These results highlight a more adaptive and efficient alternative to conventional low-rank tuning by automatically selecting the most relevant ranks based on training data. Crucially, by amplifying important ranks through learned higher importance weights, our approach significantly enhances downstream adaptation.

## B.7    Parameter-space Directional Analysis

To further investigate how CoDyRA structures its parameter updates across sequential tasks, we conduct a parameter-space directional analysis following prior work on directional interference in LoRA-based continual learning. Specifically, we compute the cosine similarity between the LoRA **A**-matrix updates obtained after each task, comparing every task's update direction against those from all other tasks. This metric captures the degree to which updates for different tasks align or interfere: higher similarity indicates that tasks induce overlapping update directions, while lower similarity reflects more orthogonal and thus less interfering modifications. As shown in Figure 18, CoDyRA consistently produces lower cross-task similarity than a baseline that trains a new LoRA adapter per task and merges them without additional constraints. Despite imposing no explicit orthogonality regularization, CoDyRA naturally encourages more decorrelated update directions, revealing an inherent structural property that helps mitigate destructive interference between tasks.

