# OpenReview forum: "Adaptive Rank, Reduced Forgetting: Knowledge Retention in Continual Learning Vision-Language Models with Dynamic Rank-Selective LoRA"
_ICLR.cc/2026/Conference — Submitted to ICLR 2026_

### Official Review · Reviewer_P5E9 · 2025-10-30

**Soundness:** 2
**Presentation:** 3
**Contribution:** 2
**Rating:** 4
**Confidence:** 3

**Summary:**

This paper addresses the challenge of catastrophic forgetting in continual learning (CL) for pre-trained vision-language models (VLMs) like CLIP. The authors note that existing methods have significant drawbacks: approaches relying on task-specific modules add inference complexity and require task-ID prediction, while replay-based methods incur high storage and computational costs.
To overcome these limitations, this paper explores performing CL through Low-Rank Adaptation (LoRA), a natural and efficient update mechanism. The authors first systematically analyze how LoRA's rank and placement affect the trade-off between "plasticity" (the ability to learn new tasks) and "stability" (the ability to retain old knowledge). Their key finding is that a relatively higher rank promotes plasticity but exacerbates forgetting, while a relatively lower rank enhances stability but limits the model's adaptation. They find that an optimal balance exists at a moderately small rank, although this balance point varies across different parameter locations and tasks.
Based on this analysis, the paper proposes CoDyRA (Continual Dynamic Rank-Selective LoRA). This method continually updates the VLM by adaptively optimizing the rank of each LoRA adapter. CoDyRA achieves this by jointly optimizing two objectives: (1) the standard new-task learning objective, which drives plasticity; and (2) a sparsity-promoting $l_1$ regularization applied to a set of learnable "importance weights" for each rank. This regularization dynamically minimizes the number of active ranks, forcing the model update to remain closer to its previous state, thereby reducing interference and forgetting.
The main contributions of this paper include:
A systematic study of the impact of LoRA's rank and placement on the plasticity-stability trade-off in VLM continual learning.
The proposal of the CoDyRA method, which uses sparsity-promoting regularization to adaptively select and minimize the rank of LoRA. It operates without storing past data, requiring task information, or adding task-specific components.

**Strengths:**

1. Solid Analysis and Clear Motivation: A core strength of this paper is the systematic analysis provided in Section 3.2. The authors delve into the impact of the rank and placement of LoRA on the tradeoff between plasticity (learning new knowledge) and stability (retaining old knowledge). This analysis clearly reveals why a fixed-rank LoRA strategy is not optimal and provides strong motivation and design guidance for the subsequently proposed adaptive-rank method.

2. Ingenious Parameter Update Mechanism: CoDyRA does not directly optimize the discrete "rank," but instead introduces a set of learnable "importance weights" ($w^{t,m}$) for each LoRA module and innovatively combines them with $l_1$ sparse regularization. This design cleverly transforms the difficult discrete rank selection problem into a solvable continuous sparse optimization problem. Furthermore, the authors employ the proximal gradient method and its soft thresholding operation to effectively solve the non-differentiability optimization problem caused by the $l_1$ norm.

3. The paper provides robust ablation experiments and parameter sensitivity analysis (Section 4.4) 9. The authors not only verified the influence of LoRA insertion position and initial rank 10, but also conducted in-depth analysis of key hyperparameters, such as the maximum pruning threshold ($\kappa_{max}$) and dense-iteration ratio 11, fully demonstrating the rationality and robustness of the model design.

**Weaknesses:**

The paper's core assumption is that "minimizing LoRA rank" can serve as an effective proxy for "reducing catastrophic forgetting". However, the analysis in Section 3.2 (Fig. 3) primarily demonstrates a correlation between rank and forgetting, but fails to deeply investigate the causality. The root cause of forgetting is the interference of the parameter update direction with old tasks, not just the rank of the update. For instance, a high-rank update might not cause forgetting if its update direction is orthogonal to the parameter subspace of old tasks. Conversely, a low-rank (or even rank-1) update could be catastrophic if its direction is incorrect (e.g., directly opposes a critical gradient direction for a previous task). The paper lacks a deeper parameter-space analysis to substantiate this assumption. For example, does the low-rank increment $\Delta W$ generated by CoDyRA's $l_1$ sparsification truly interfere less with old knowledge in terms of its update direction compared to standard LoRA? A specific analysis of this direction's orthogonality or interference is missing. Currently, the validity of "low rank" as a robust proxy for "low forgetting" has not been sufficiently theoretically or empirically demonstrated.

**Questions:**

### Potential Typo in Core Equations (Eq. 4 & 7)

There appears to be a typo in the soft-thresholding operator used for the \( l_1 \) regularization.

1. **The Goal:**
   The paper aims to use \( l_1 \) regularization to promote sparsity by pushing the importance weights \( w \) **toward 0**.

2. **The Formula:**
   However, Eq. (4) and Eq. (7) define the operator with a **plus sign**:

   \[
   w_{i} := \mathbb{I}(|\hat{w}_{i}| > \kappa) \cdot (\hat{w}_{i} + \operatorname{sign}(\hat{w}_{i}) \cdot \kappa)
   \]

3. **The Contradiction:**
   This formula would actually *amplify* the weights (e.g., 5 becomes 6), moving them **away from 0**.
   This is the opposite of sparsity.

---

Should this formula use a **minus sign**
(\( \ldots - \operatorname{sign}(\hat{w}_{i}) \cdot \kappa \))
to correctly implement the soft-thresholding (shrinkage) operation?
Please clarify if this is a typo in the manuscript and if the correct operator was used in the implementation.

---

> ### Author Response · Authors · 2025-12-04
> **Thank you for your comments! (Part 1/2)**
>
> **W1 (a). Connections between rank and forgetting.**
>
> We thank the reviewer for this insightful comparison.
>
> A low rank $r$ acts as a structural regularizer, limiting the update $\Delta W$ to the minimal dimension necessary. This minimizes the update norm $||\Delta W||_F$ and inherently reduces the probability of conflicting with the critical subspaces of prior tasks.
>
> While both Orthogonal LoRA and CoDyRA can be seen as aiming to mitigate Subspace Interference, they operate on fundamentally different principles with distinct trade-offs. They work on different ways focusing on different perspectives. And they can also be complenatry with each other.
>
> Here, we highlight three specific benefits of CoDyRA over direct orthogonal regularization in our setting.
>
> 1. Orthogonal regularization rigidly constrains the new update $\Delta W_{new}$ to be perpendicular to previous subspaces ($\Delta W_{old}$). While this prevents interference, it applies a strong restriction on learning capability, tending to prevent positive transfer. If a new task shares semantic features with a previous one (e.g., effectively refining a "fur detector" in the vision encoder), orthogonal methods force the model to learn a redundant, separate feature vector. CoDyRA, by regularizing for simplicity (rank sparsity) rather than geometry (angle), allows the model to align with previous knowledge when beneficial, while minimizing the footprint of the update to prevent accidental overwriting of unrelated knowledge. On the other hand, orthogonal regularization methods typically need to be relaxed during training to allow sufficient plasticity, meaning that forgetting can still occur even when orthogonality is enforced. In contrast, CoDyRA achieves this balance in a more direct and effective way.
>
> 2. Explicit orthogonal regularization typically requires storing the projection matrices (singular vectors) of previous tasks to calculate the orthogonality loss (e.g., $||\Delta W_{new}^T \Delta W_{old}||_F$). This introduces memory overhead and dependency on past task statistics. CoDyRA reduces interference implicitly via the $l_1$ sparsity on rank importance. This is a strictly local objective that requires no memory of previous tasks, aligning with our goal of a natural, memory-free PTM update.
>
> 3. Standard orthogonal methods often assume a fixed rank and try to rotate it. If the fixed rank is higher than the task's intrinsic dimension, the "extra" dimensions act as noise vectors that drift the weights unnecessarily. CoDyRA actively prunes these dimensions. A rank that is pruned to zero has, by definition, zero interference with previous tasks. Thus, CoDyRA achieves "perfect orthogonality" (zero overlap) for irrelevant dimensions, while allowing flexible updates for relevant ones.
>
> ---
>
> **W1 (b). Analysis of parameter orthogonality.**
>
> In the revised manuscript, we include a parameter-space analysis (**Figure 18 in the Appendix**) examining the inherent directional effects of CoDyRA's updates between sequential tasks. We compare CoDyRA against a simple baseline that learns and merges a new LoRA adapter for each task. Directional interference is quantified using the cosine similarity between the LoRA matrix $A$ (following InfLoRA and O-LoRA) components across tasks. We observe that, despite CoDyRA imposing no explicit orthogonal regularization or constraints, it inherently obtains a lower cosine similarity, indicating better implicit orthogonality of the parameter updates. Moreover, complementing orthogonal regularization techniques with our rank-wise importance control presents a promising direction for future research.

---

> ### Author Response · Authors · 2025-12-04
> **Thank you for your comments! (Part 2/2)**
>
> **Q1. Typo in Eq. 4 and Eq. 7**
> We sincerely thank the reviewer for their meticulous reading and for identifying this typo. We confirm that the use of the plus sign ($+$) in Eq. 4 and Eq. 7 is indeed a typo in the manuscript; the minus sign ($-$) is the correct operator for soft-thresholding to induce sparsity. Crucially, we confirm that our code implementation is correct. We utilized the standard soft-thresholding (shrinkage) operation in all our experiments. As illustrated in the snippet below from our codebase, we correctly shrink the weights toward zero:
> ```
> # Example logic from our soft-thresholding implementation:
> # 1. Shrink positive weights toward 0
> importance_weights[importance_weights > threshold] -= threshold
> # 2. Shrink negative weights toward 0
> importance_weights[importance_weights < -threshold] += threshold
> # 3. Zero out weights within the threshold
> importance_weights[importance_weights.abs() < threshold] = 0.0
> ```
> We have corrected the equations in the revised manuscript to properly reflect the subtraction operation ($x - \text{sign}(x) \cdot \kappa$).
>
> ---
>
> We sincerely thank the reviewer for the insightful comments and discussion. In response, we have included additional discussions and experiments in the revision (highlighted in blue) to address the concerns. We hope that these updates help alleviate the issues raised and kindly ask that you consider revising the score if you find them satisfactory.

---

### Official Review · Reviewer_oHrv · 2025-10-31

**Soundness:** 2
**Presentation:** 4
**Contribution:** 2
**Rating:** 4
**Confidence:** 4

**Summary:**

This paper proposes Continual Dynamic Rank-Selective LoRA (CoDyRA) for continual learning, mainly for cross-domain continual learning. The proposed method has been developed based on CLIP VLM and LoRA approach. The paper is well-presented and comprehensive. The paper is easy to follow with helpful highlights for the reader. Despite its positive values, this paper has a few fundamental issues; please see the weaknesses.

**Strengths:**

Strength:
(1). This paper discusses an interesting up-to-date sub-problem of CL, i.e, cross-domain CL.

(2). The motivation of the proposed method is clear.

(3). The writing is well presented with helpful highlights, charts, and diagrams.

(4). The experiment results are comprehensively delivered and conducted on many datasets.

**Weaknesses:**

(1). The main idea of the proposed method is adding new trainable weight w^{t,m}_i associated with B^{t,m}_{:i}. The proposed trainable weights are supposed to make \delta W^{t,m} more adaptive. From the methodology perspective, the idea is arguably not novel as the idea of trainable weights was already applied in previous methods, e.g, ConvPrompt [CVPR 2024].

(2). No theoretical and numerical proofs showing that the trainable weights idea improves model adaptation and reduces model forgetting significantly.

(3). Performance issue: Even though the proposed method achieves the highest performance on average, it is significantly outperformed by the previous method (RAIL-Primal), i.e, 11\% on Caltect101 dataset and 3.9\% on Flowers dataset.

(4). Continual learning is the art of defying catastrophic forgetting (CF). But, I do not see a comprehensive forgetting analysis.

(5). The paper misses a comparison of the proposed method to the newest CLIP and LoRA-based CL methods, i.e, CLAP4CLIP (NeurIPS 2024), C-CLIP (ICLR 2025), Mind-the-Gap (ICCV 2025), CL-LoRA(CVPR-2025), InfLORA (CVPR 2024).


References:

[1]. CLAP4CLIP: Continual learning with probabilistic finetuning for vision-language models.

[2]. C-CLIP: Multimodal continual learning for vision-language model.

[3]. Mind the gap: Preserving and compensating for the modality gap in clip-based continual learning.

[4]. CL-LoRA: Continual Low-Rank Adaptation for Rehearsal-Free Class-Incremental Learning

[5]. Inflora: Interference-free low-rank adaptation for continual learning.

[6] Convolutional prompting meets language models for continual learning (ConvPrompt)

**Questions:**

Please address the weaknesses.

---

> ### Author Response · Authors · 2025-12-04
> **Thank you for your comments! (Part 1/4)**
>
> We thank the reviewer for the insightful and detailed feedback, particularly acknowledging the importance of our focus on the interesting, up-to-date sub-problem of cross-domain Continual Learning. We appreciate the comments on the clarity of our method's motivation, the high quality of the writing, and the comprehensive experiments across many datasets. We now address the remaining concerns point by point.
>
> ---
>
> **W1. Discussions regarding ConvPrompt [1].**
> We thank the reviewer for the reference. We respectfully clarify that our sparse rank-selective weighting is **fundamentally different** and **independent** from ConvPrompt’s input-level conditional prompt selection. Although both involve the "idea of trainable weights"—a very common strategy in deep learning—the mechanisms and purposes here are entirely distinct. Thus, we believe the critique of novelty based on ConvPrompt is unfounded.
>
> We have included a detailed discussion of these distinctions in **Appendix A.3** and summarize them below:
>
> **(a) Fundamental mechanism:**
> - **ConvPrompt (Input-conditioned weights)**: Its weights are dynamically calculated at inference time by conditioning on the input features (e.g., using [CLS] embeddings to query prompt keys via cosine similarity). These weights change for every single image to perform instance-aware prompt combination.
> - **CoDyRA (Input-Independent weights)**: Our importance weights are global learnable parameters optimized directly via gradient descent and soft thresholding. They are not conditioned on input features. Instead, they serve as a importance score for each rank of updates, determining the effective rank of the update matrix $\Delta W$.
>
> **(b) Usage**: Because ConvPrompt relies on input conditioning, it must compute these weights on-the-fly for every sample, introducing inference overhead. In contrast, CoDyRA's weights are structural parameters that are merged into the linear weights after training. This results in zero inference overhead, maintaining the original efficiency of the pre-trained backbone.
>
> **\(c) Functionality**: ConvPrompt’s weights act as an attention mechanism for input adaptation. CoDyRA’s weights, guided by $l_1$ sparsity, act as a capacity control mechanism to balance plasticity and stability by pruning redundant ranks. Thus, CoDyRA represents a method for parameter-efficient fine-tuning, which is technically and functionally distinct from the conditional prompting mechanism in ConvPrompt.
>
> [1] Roy et al., Convolutional prompting meets language models for continual learning, CVPR, 2024.
>
> ---
>
> **W2. Theoretical and numerical proofs for the trainable weights mechanism.**
>
> We respectfully clarify that the effectiveness of our trainable weights mechanism is substantiated by both rigorous numerical proof and theoretical grounding:
>
> **(a) Numerical proof:** In our submission, We have explicitly isolated the contribution of the trainable weights mechanism in **Table 6 (Appendix B.1)**. We compared CoDyRA against standard fixed-rank LoRA baselines ($r=4, 8, 16$). The results provide clear numerical evidence:
> - **Reduction in forgetting**: CoDyRA achieves a significantly higher "Last" accuracy (79.2%) compared to fixed-rank LoRA at $r=16$ (59.4%). This large gap empirically proves that the adaptive rank selection (driven by our trainable weights) successfully mitigates catastrophic forgetting where fixed ranks fail.
> - **Visual verification:** In **Fig. 9 (Appendix B.3)**, we visualize the "Amplification Factor." The results show that CoDyRA produces significantly sparser updates than vanilla LoRA, confirming that our mechanism effectively suppresses updates to non-essential weights, thereby reducing interference with pre-trained knowledge.
>
> **(b) Theoretical Grounding**: The trainable weights mechanism solves the plasticity-stability dilemma by dynamically optimizing the intrinsic dimensionality of the parameter update:
>  - **Minimal Sufficient Perturbation**: Rank determines the degrees of freedom. While high rank maximizes informational capacity (plasticity), it also maximizes disruption to the pre-trained manifold (instability). Our trainable weights act as a structural regularizer, enforcing the "Minimal Sufficient Perturbation" necessary to solve the current task. By minimizing the effective rank and Frobenius norm $||\Delta W||_F$, we statistically reduce the probability of the update interfering with the critical subspaces of prior tasks.
>  - **Adaptive Optimization**: Different tasks and layers possess different intrinsic dimensions. CoDyRA handles this functional variance by allocating higher ranks only to layers with high gradient influence (e.g., specific MLPs) while pruning less critical layers. This prevents the over-allocation of capacity that leads to forgetting, a nuance that fixed-rank methods fail to capture.

---

> ### Author Response · Authors · 2025-12-04
> **Thank you for your comments! (Part 2/4)**
>
> **W3. Performace Improvement of Table 1.**
>
> We appreciate the reviewer's detailed examination of the dataset-specific results.
> While we acknowledge that our performances are not always the best on all the sub-datasets, such as Food and Caltech101, we respectfully clarify that this does not constitute a general performance failure.
>
> Under the most important “Last” setting (i.e., the final performance after the full continual learning sequence), CoDyRA actually outperforms RAIL-Primal$\dagger$ on **7 out of 9** comparable datasets, indicating that our method is consistently stronger in the primary continual-learning metric.
>
> The few “non-best” cases (those lower than RAIL) noted by the reviewer is relevant to pertubations or speciality related to the dataset, such as "Food", which has different distributions comparing to others. So the performances of different methods on some specific domains/subdatgassets tend to be pertubate. For example, the best performing methods on Food changes (RAIL as mentioned by reviewer also experiences a performance drop). But, obviously, our method can achive a more stable and strong average performances and outperforms others on the dominante numnber of any one of  other resutls.
>
> The gaps highlighted by the reviewer are offset by our significant gains on other difficult domains. For example, on EuroSAT, we outperform RAIL by **+6.6\%** (**93.0\%** vs **86.4\%**), and on MNIST by **+4.1\%**.
>
> The goal of X-TAIL is robust average performance across diverse domains.
> Our method achieves the highest average accuracy (**80.9\%** vs **79.1\%**), which is the standard aggregate criterion in continual learning. This supports our claim that CoDyRA offers the best overall generalization and robustness, even if it is not always the top method on every single dataset.
>
> ---
>
> **W4. Discussions of forgetting.**
>
> We thank the reviewer for the thoughtful review. We have added a dedicated Forgetting Analysis in the revision (**Table 9**), quantifying forgetting via Backward Transfer (BWT). It reports BWT of CoDyRA and some relavent CL methods with representation udpating.
>
> Fixed-rank LoRA serves as a baseline and exhibits significant forgetting due to unconstrained updates without regularization. ZSCL relies on the replay of reference data to help mitigate forgetting. MoE-Adapters isolates task-wise knowledge within specific adapters and adopts domain prediction to further reduce interference. InfLoRA constrains parameter updates to a direction orthogonal to prior tasks to reduce interference. CoDyRA effectively mitigates forgetting by minimizing the rank of updates via sparsity regularization. This ensures the model learns effectively while remaining close to its original state, demonstrating strong retention capabilities with a BWT of only **1.87%**.
>
> Moreover, in **Table 8** of the revision, we validate our method in the language domain by continually training the LLaMA-3.2-1B-Instruct model. The low BWT scores observed in this setting further support our claims of reduced forgetting across architectures.
>
>
> **Table 9 in revision.** Forgetting analysis. We report Backward Transfer (BWT) averaged across all X-TAIL tasks, where lower values indicate better retention.
>
> | Methods | ZSCL | MoE-Adapter | InfLoRA | LoRA (r=4) | LoRA (r=8) | LoRA (r=16) | CoDyRA |
> |---|:---:|:---:|:---:|:---:|:---:|:---:|:---:|
> | BWT (%) | 8.78 | 5.43 | 3.12 | 13.94 | 14.24 | 14.86 | 1.87 |

---

> ### Author Response · Authors · 2025-12-04
> **Thank you for your comments! (Part 3/4)**
>
> **W5. Extended comparisons and discussions of related works.**
>
> We thank the reviewer for these valuable suggestions and have expanded our discussion of all mentioned related works in the revised **Appendix A.3**. Furthermore, during the rebuttal period, we reproduced two relevant LoRA-based methods (CL-LoRA and InfLoRA) within our experimental setting (**Responses Part 4/4**)to provide a rigorous quantitative comparison. Regarding some other Class-Incremental Learning (CIL) methods, we note that a direct quantitative comparison is not feasible within our zero-shot transfer setting. For instance, methods like CLAP4CLIP rely on memory replay buffers and specifically designed adapters that do not support inference on unseen classes or domains.
>
> (a) CLAP4CLIP [2]. This framework uses probabilistic variational inference to model task-specific distributions, which introduces additional complexity from probabilistic modeling and sampling. CoDyRA enforces structural sparsity directly in the parameter space, yielding a more stable and computationally efficient optimization procedure.
>
> (b) C-CLIP [3]. This method utilizes an auxiliary Contrastive Knowledge Consolidation (CKC) objective function to preserve zero-shot performance. Unlike C-CLIP, which relies on external loss constraints to enforce retention, CoDyRA relies on intrinsic parameter regulation ($l_1$ sparsity).
>
> \(c) MG-CLIP (Mind the Gap) [4]. This method adds auxiliary visual-space classifiers at inference time to mitigate the modality gap, altering the standard inference pipeline. CoDyRA preserves the original CLIP architecture and deployment, maintaining vision–language alignment by regulating the update rank, with no extra inference overhead or external modules.
>
> (d) CL-LoRA [5]. This method adopts a dual-adapter design with separate task-shared and task-specific modules plus gradient reassignment, leading to higher architectural and parameter-management complexity. CoDyRA is significantly simpler, using a single unified adapter per layer and achieving the plasticity–stability trade off via dynamic rank selection (pruning non-essential ranks) rather than architectural bifurcation.
>
> (e) InfLoRA [6]: This method projects updates into a subspace strictly orthogonal to prior tasks to eliminate interference. This enforces a "hard" constraint that can overly restrict the solution space and limit plasticity. CoDyRA applies a "soft" constraint via sparsity-promoting regularization.
>
> [2] Jha et al., CLAP4CLIP: Continual Learning with Probabilistic Finetuning for Vision-Language Models, NeurIPS, 2024.
>
> [3] Liu et al., C-CLIP: Multimodal Continual Learning for Vision-Language Model, ICLR, 2025.
>
> [4] Huang et al., Mind the Gap: Preserving and Compensating for the Modality Gap in CLIP-Based Continual Learning, arXiv, 2025.
>
> [5] He et al., CL-LoRA: Continual Low-Rank Adaptation for Rehearsal-Free Class-Incremental Learning, CVPR, 2025.
>
> [6] Liang et al., InfLoRA: Interference-Free Low-Rank Adaptation for Continual Learning, CVPR, 2024.
>
> ---
>
> We sincerely thank the reviewer for their insightful comments and constructive feedback. In response, we have incorporated additional discussions and experimental results into the revised manuscript (highlighted in blue) to address your concerns. We hope these updates satisfactorily resolve the issues raised, and we respectfully invite you to reconsider your score.

---

> ### Author Response · Authors · 2025-12-04
> **Thank you for your comments! (Part 4/4)**
>
> **Updated entries of Table 6 in Appendix.** We have added comparisons with InfLoRA and CL-LoRA.
>
> |              | Aircraft | Caltech101 |  DTD  | EuroSAT | Flowers | Food | MNIST | OxfordPet | Cars | SUN397 | Avg. |
> |-------------:|:--------:|:----------:|:-----:|:-------:|:-------:|:----:|:-----:|:---------:|:----:|:------:|:-------:|
> | Zero-Shot    | 23.5     | 76.8       | 37.3  | 36.7    | 63.6    | 84.0 | 46.7  | 86.7      | 66.1 | 63.7   | 58.5    |
> | Transfer     |          |            |       |         |         |      |       |           |      |        |         |
> | MoE-Adapter  | -        | 71.0       | 34.9  | 19.2    | 63.0    | 86.6 | 20.0  | 87.2      | 63.7 | 58.6   | 56.0    |
> | InfLoRA      | -        | 75.8       | 34.5  | 29.2    | 58.1    | 73.4 | 38.6  | 79.5      | 47.7 | 50.3   | 54.1    |
> | CL-LoRA      | -        | 73.3       | 33.7  | 29.5    | 58.5    | 80.3 | 43.1  | 85.5      | 61.5 | 59.2   | 58.3    |
> | CoDyRA       | -        | 74.3       | 36.8  | 44.2    | 69.9    | 83.5 | 42.8  | 88.9      | 64.6 | 63.4   | 63.2    |
> | Average      |          |            |       |         |         |      |       |           |      |        |         |
> | MoE-Adapter  | 43.6     | 77.9       | 52.1  | 34.7    | 75.9    | 86.3 | 45.2  | 87.4      | 66.6 | 60.2   | 63.0    |
> | InfLoRA      | 38.3     | 84.2       | 60.1  | 45.8    | 79.0    | 77.5 | 61.8  | 82.8      | 52.2 | 52.2   | 63.4    |
> | CL-LoRA      | 39.7     | 79.2       | 56.7  | 58.5    | 79.1    | 81.1 | 64.0  | 78.0      | 63.6 | 60.6   | 66.0    |
> | CoDyRA       | 41.4     | 81.0       | 58.7  | 77.8    | 83.4    | 84.6 | 64.5  | 90.4      | 67.2 | 64.4   | 71.3    |
> | Last         |          |            |       |         |         |      |       |           |      |        |         |
> | MoE-Adapter  | 43.2     | 78.7       | 57.6  | 32.8    | 79.4    | 86.0 | 86.7  | 87.8      | 78.2 | 74.2   | 70.5    |
> | InfLoRA      | 38.3     | 85.2       | 66.6  | 52.9    | 92.9    | 81.6 | 96.6  | 90.4      | 70.1 | 69.3   | 74.4    |
> | CL-LoRA      | 39.7     | 79.8       | 62.5  | 70.9    | 92.9    | 81.9 | 95.2  | 90.5      | 72.4 | 73.1   | 75.9    |
> | CoDyRA       | 37.7     | 81.5       | 65.1  | 89.9    | 91.4    | 85.5 | 96.8  | 93.3      | 77.3 | 73.5   | 79.2    |

---

### Official Review · Reviewer_VY2p · 2025-11-01

**Soundness:** 3
**Presentation:** 3
**Contribution:** 3
**Rating:** 6
**Confidence:** 3

**Summary:**

This paper targets continual learning (CL) for vision-language models like CLIP by introducing CoDyRA. This method dynamically optimizes the rank of LoRA adapters during sequential task learning to balance the plasticity and stability. The authors systematically analyze the impact of LoRA rank and placement on learning-forgetting trade-offs and propose an adaptive rank-selection mechanism driven by sparsity-promoting regularization. Extensive experiments on benchmarks such as MTIL and X-TAIL demonstrate improved performance over SOTA methods in retaining pre-trained capabilities while improving generalization, with no inference overhead. Though very intriguing and promising, this work could benefit from a more in-depth theoretical analysis and a more structured presentation.

**Strengths:**

1. It introduces a fine-grained, adaptive approach to LoRA-based CL, offering a novel perspective supported by convincing experimental validation.
2. The method is straightforward to implement and exhibits potential for scalability due to its simplicity.
3. Comprehensive experiments across diverse benchmarks and model configurations, including visualizations, robustly substantiate the core claim that rank manipulation addresses the learning-forgetting trade-off effectively.

**Weaknesses:**

1. Though the authors have provided a preliminary empirical analysis of the impact of the lora location and rank (sec 3.2), the study lacks direct theoretical derivation or proof, necessitating deeper analytical foundations beyond empirical results.
2. Marginal improvements in Tables 1 and 2 (often fractions of a percent, less than 1%) raise concerns about the method’s effectiveness and generality compared to state-of-the-art approaches.
3. Most experiments use the ViT-B/16 backbone of CLIP. More tests on a larger or different model architecture and different pre-trained parameters could give a broader impact assessment.

**Questions:**

See the weakness part.

---

> ### Author Response · Authors · 2025-12-04
> **Thank you for your comments! (Part 1/2)**
>
> We thank the reviewer for the positive feedback, particularly acknowledging the novelty of our fine-grained, adaptive approach to LoRA-based CL, the method's simplicity and potential for scalability, and the comprehensive experimental validation and visualizations, which robustly substantiates our core claims. We now address the remaining concerns point by point.
>
> ---
>
> **W1. Theoretical foundations of our analyses in Section 3.2.**
>
> We thank the reviewer for highlighting the need for deeper analytical foundations. While a formal theoretical proof of the non-linear dynamics of LoRA in a continual learning setting remains an open problem in the field, we believe our empirical results of the 'Plasticity-Stability' trade-off in relation to Rank and Module Placement provides a critical guide for CL design. The detailed analyses and experiments on CL tasks further demonstrate the value and effectiveness of the proposed methods. To address the reviewer's concern, we provide the following theoretical interpretations that ground our three takeaways (**Appendix A.2** in revision).
>
> **1. On Applying LoRA to All Modules (Takeaway 1)**.
>
> The original LoRA paper applies LoRA only to selected modules (mainly the attention components). Many subsequent LoRA and LoRA-based continual learning works (e.g., InfoLoRA, MoE-Adapters, O-LoRA, etc.) also apply LoRA to only part of the Transformer modules, though the specific choices vary. To clarify this design space, we conducted an analysis to determine which module configuration is most effective. Our results show that applying LoRA to all layers yields clear benefits.
>
> While the theoretical analysis of standard Transformers and full LoRA training dynamics is complex, the empirical Neural Tangent Kernel (eNTK) [1] offers a practical framework for theoretical investigation. The necessity of applying LoRA to all layers, rather than just attention modules, can be explained through the lens of the eNTK .
>
> The eNTK of LoRA approximates that of Full Fine-Tuning (FullFT) only when LoRA is applied to the layers that contain the majority of the parameters. Since the kernel is based on the dot products of gradients ($K(i, j) = g_i \cdot g_j$), layers with more parameters (e.g., MLPs) typically exert the most influence on the kernel. Restricting updates to attention layers limits the model's ability to approximate the learning dynamics of FullFT, resulting in "slower learning" or reduced plasticity.
>
> **2. On the Rank-Dependent Plasticity-Stability Balance (Takeaway 2).**
>
> We formalize the causality of this balance by defining the trade-off as an informational dilemma rooted in the subspace dimensionality:
>
> **High Rank $\rightarrow$ High Plasticity / Low Stability**: The rank ($r$) dictates the intrinsic dimensionality (degrees of freedom) of the update subspace $\Delta W$. A high rank maximizes the informational capacity of the parameter shift. In the NTK context, maximizing rank pushes the update closer to the FullFT regime. While this allows for rapid acquisition of new task knowledge (High Plasticity), the larger informational perturbation maximizes disruption to the pre-trained knowledge manifold, leading to catastrophic forgetting (Low Stability).
>
> **Low Rank $\rightarrow$ High Stability / Low Plasticity**: Conversely, restricting rank constrains the degrees of freedom. This forces the model to encode only the minimal, essential information for the new task. This minimization of total informational perturbation ensures the model remains informationally proximal to its previous state—a necessary condition for stability.
>
> A lower rank $r$ of the LoRA implies a smaller dimension for the update $\Delta W$. A low rank acts as a structural regularizer. It forces the update to be the "simplest" change necessary to solve the current task. This minimizes the norm $||\Delta W||_F$ and reduces the probability that the update vector on the critical subspace of previous tasks.
>
> **3. On the Necessity of Adaptive Optimization (Takeaway 3).**
>
> The need for adaptive rank selection stems from the functional variance of model components and the different information requirements of tasks.
>
> Following the theory on intrinsic dimension of objective landscapes [2], different tasks require different degrees of freedom to be solved. And, not all layers contribute equally to the objectives (eNTK or feature transformation in eNTK theory). Adaptivity allows the model to allocate higher ranks to layers with high gradient influence (e.g., specific MLPs) while enforcing sparsity on less critical layers.
>
> [1] Malladi et al., A Kernel-Based View of Language Model Fine-Tuning, ICML, 2023.
>
> [2] Li et al., Measuring the intrinsic dimension of objective landscapes. ICLR, 2018.

---

> ### Author Response · Authors · 2025-12-04
> **Thank you for your comments! (Part 2/2)**
>
> **W2. Improvements of Tables 1 and 2.**
>
> **(1) Continual learning performance of Table 1**:
> We respectfully note that the seemingly small gains in **Table 1** should be interpreted in light of the strength and design of the competing methods. When interpreted in context, our results represent a qualitative breakthrough:
> - **(a) Breaking the zero-shot "ceiling" (Transfer Accuracy)**: In Continual Learning with Foundation Models, the primary challenge is integrating new knowledge without destroying pre-trained generalization. As shown in **Table 1**, competitive baselines (e.g., MoE-Adapters, RAIL) are structurally **upper-bounded** by the frozen backbone for zero-shot prediction, **capping** their performance at the **original CLIP** baseline. CoDyRA achieves 63.2%, being **the only method** that surpasses the zero-shot predictions of the pre-trained model. This proves CoDyRA effectively consolidates knowledge to upgrade the model, rather than merely overfitting to seen tasks.
> - **(b) Consistent and robust gains (Last Accuracy)**: CoDyRA demonstrates remarkably consistent gains. Under the strict constraint of **continually updating the foundation model without auxiliary modules**, CoDyRA significantly outperforms the fixed-rank LoRA baseline (**Table 5**) by a margin of around **20%**. Furthermore, compared to prior works that **rely on additional modules**, CoDyRA establishes a new state-of-the-art: it shows consistent gains over representation learning methods (e.g., ZSCL, MoE-Adapter)  and robust improvements over the regression-head method RAIL on most of the sub-tasks in continual learning.
> - **\(c) Efficiency**: CoDyRA achieves these SOTA results using only 4.4M trainable parameters (**Table 4**). Compared to MoE-Adapter, we outperform MoE while using much fewer parameters (4.4M vs. 59.8M). Compared to RAIL, we achieve higher accuracy without the need for memory banks or high-dimensional projectors. CoDyRA achieves higher accuracy with much less training complexity with **zero inference overhead**.
> - **(d) Statistical robustness**: **Table 8** confirms that these gains are not noise. The low standard deviation across three runs confirms that CoDyRA's improvements are stable and statistically significant.
>
> **(2) Improved generalizations in Table 2**:
> **Table 2** presents the performance on unseen data after continual learning. We emphasize that CoDyRA is the **only method** capable of **breaking the zero-shot performance ceiling** of the pre-trained model.
>
> Continually updating a pre-trained model typically causes catastrophic forgetting (as shown in **Fig. 1**). To avoid this, prior methods (e.g., MoE-Adapters, RAIL) isolate new task learning into separate parameters and rely on routing or gating mechanisms. As discussed in our Introduction, these methods introduce additional components that prevent **leveraging new knowledge** acquired during fine-tuning to improve generalization.
>
> As shown in **Table 2**, these methods collapse to the baseline performance for unseen data. Because they rely on domain prediction, they must default to the original pre-trained model for unseen inputs to avoid negative transfer from modules trained on prior tasks. Consequently, they are upper-bounded by the frozen backbone and cannot leverage new knowledge acquired during fine-tuning to improve generalization.
>
> In contrast, CoDyRA aims to unleash the continual representation capability of the backbone itself via adaptive-rank LoRA. By merging optimized-rank updates back into the pre-trained weights, CoDyRA continually updates the model with knowledge of prior tasks. This enables the backbone to directly leverage knowledge accumulated from previous tasks for inference on unseen data.
>
> Surpassing the inherent zero-shot ceiling of the base model by 1.76% (as evidenced by our superior Transfer Accuracy in **Table 2**) represents a significant breakthrough in continual CLIP adaptation. It proves that CoDyRA effectively integrates new knowledge to upgrade the model's generalizable representations, going beyond the mere preservation of pre-trained knowledge.
>
> ---
>
> **W3. Extention to other backbones.**
> Thanks for your suggestions! To demonstrate the generalization of our method, we have evaluated CoDyRA using a different VLM backbone, BLIP, as detailed in Appendix B.5 (**Table 11**). Moreover, in **Table 8** of the revision, we extend our evaluation to the language domain by continually training the LLaMA-3.2 model. In both settings, CoDyRA consistently outperforms prior works with reduced forgetting, demonstrating its general effectiveness across diverse foundation models.
>
> ---
>
> We sincerely thank the reviewer for the insightful comments and discussion. In response, we have included additional discussions and experiments in the revision (highlighted in blue) to address the concerns. We hope that these updates help alleviate the issues raised and kindly ask that you consider revising the score if you find them satisfactory.

---

### Official Review · Reviewer_TRej · 2025-11-06

**Soundness:** 4
**Presentation:** 3
**Contribution:** 3
**Rating:** 6
**Confidence:** 4

**Summary:**

This paper proposes CoDyRA (Continual Dynamic Rank-Selective LoRA), a continual learning method for vision–language models like CLIP that updates pre-trained models using LoRA adapters with adaptively optimized ranks. By balancing plasticity and stability through dynamic rank selection and sparsity regularization, CoDyRA enables efficient continual updates without replay, task-specific modules, or added inference cost, achieving state-of-the-art performance while preserving prior knowledge.

**Strengths:**

1. The paper is well-structured and clearly written, making it easy to follow.

2. The study tackles an important problem in continual learning by employing low-rank adaptation.

3. The experimental evaluation is comprehensive, incorporating a wide range of baselines and datasets, which enhances the credibility of the paper’s conclusions.

**Weaknesses:**

1. Some related works appear to have been overlooked. There are also several recent studies that attempt to adjust their architectures dynamically in continual learning, such as TreeLoRA.

    TreeLoRA: Efficient Continual Learning via Layer-Wise LoRAs Guided by a Hierarchical Gradient-Similarity Tree.

2. Can the authors extend the proposed method to large language models to further validate its scalability?

**Questions:**

See weaknesses above.

---

> ### Author Response · Authors · 2025-12-04
> **Thank you for your comments!**
>
> We thank the reviewer for the positive feedback on the clarity of the paper, the importance of the problem, and the comprehensiveness of our experiments. We now address the remaining concerns point by point.
>
> ---
>
> **W1. Comparisons to TreeLoRA [1].**
> We thank the reviewer for pointing out TreeLoRA. TreeLoRA mitigates interference by structurally expanding the model and isolating task-specific knowledge in separate branches, where routing decisions rely on accurate similarity estimation. To provide a comprehensive comparison, we have included TreeLoRA results in the **updated Table 6** (X-TAIL with CLIP) and **Table 8** (TRACE with LLaMA-3.2) of the revised manuscript.
>
> While such kind of isolation-based design is effective and commonly studied for continual learning, its focus and perspective differ fundamentally from the paradiam in our method. In contrast, CoDyRA supports continual learning within a unified model update process, focusing on alleviating forgetting during model updates by adapting and minimizing the LoRA rank through sparsity-regularized rank-importance weights, rather than relying on structural expansion or task-specific branches.
>
> **Updated entries of Table 6 in Appendix.** We have added comparisons with TreeLoRA.
>
> |             | Aircraft | Caltech101 |  DTD  | EuroSAT | Flowers | Food | MNIST | OxfordPet | Cars | SUN397 | Average |
> |------------|:--------:|:----------:|:-----:|:-------:|:-------:|:----:|:-----:|:---------:|:----:|:------:|:-------:|
> | Zero-Shot  | 23.5     | 76.8       | 37.3  | 36.7    | 63.6    | 84.0 | 46.7  | 86.7      | 66.1 | 63.7   | 58.5    |
> | Transfer   |          |            |       |         |         |      |       |           |      |        |         |
> | MoE-Adapter| -        | 71.0       | 34.9  | 19.2    | 63.0    | 86.6 | 20.0  | 87.2      | 63.7 | 58.6   | 56.0    |
> | TreeLoRA   | -        | 76.0       | 36.3  | 34.0    | 58.5    | 77.2 | 43.3  | 82.2      | 49.8 | 55.8   | 57.0    |
> | CoDyRA     | -        | 74.3       | 36.8  | 44.2    | 69.9    | 83.5 | 42.8  | 88.9      | 64.6 | 63.4   | 63.2    |
> | Average    |          |            |       |         |         |      |       |           |      |        |         |
> | MoE-Adapter| 43.6     | 77.9       | 52.1  | 34.7    | 75.9    | 86.3 | 45.2  | 87.4      | 66.6 | 60.2   | 63.0    |
> | TreeLoRA   | 19.3     | 81.4       | 55.3  | 63.9    | 77.6    | 80.4 | 64.3  | 85.0      | 54.8 | 57.6   | 64.0    |
> | CoDyRA     | 41.4     | 81.0       | 58.7  | 77.8    | 83.4    | 84.6 | 64.5  | 90.4      | 67.2 | 64.4   | 71.3    |
> | Last       |          |            |       |         |         |      |       |           |      |        |         |
> | MoE-Adapter| 43.2     | 78.7       | 57.6  | 32.8    | 79.4    | 86.0 | 86.7  | 87.8      | 78.2 | 74.2   | 70.5    |
> | TreeLoRA   | 16.0     | 79.1       | 59.2  | 62.3    | 83.0    | 83.6 | 95.0  | 89.9      | 71.9 | 74.2   | 71.4    |
> | CoDyRA     | 37.7     | 81.5       | 65.1  | 89.9    | 91.4    | 85.5 | 96.8  | 93.3      | 77.3 | 73.5   | 79.2    |
>
>
> [1] Qian et al., TreeLoRA: Efficient Continual Learning via Layer-Wise LoRAs Guided by a Hierarchical Gradient-Similarity Tree, ICML, 2025.
>
> ---
>
> **W2. Implementation and experiments on LLMs**
>
> We thank the reviewer for this constructive suggestion! In the revised manuscript, we have extended the evaluation of CoDyRA to Large Language Models (LLMs), adopting the experimental setup used in TreeLoRA. Specifically, we evaluated LLaMA-3.2-1B-Instruct on the TRACE [2] benchmark, reporting Overall Performance (OP) and Backward Transfer (BWT). The results, presented in **Table 8**, demonstrate that CoDyRA scales effectively to LLM architectures and achieves performance competitive with prior works.
>
> **Table 8 in revison.** Comparison with a broad range of CL methods on the TRACE benchmark using the LLaMA-3.2-1B-Instruct backbone. We report Overall Performance (OP (\%) $\uparrow$) and Backward Transfer (BWT (\%) $\downarrow$).
>
> |     | FIX(ICL) | SeqLoRA |  OGD  |  GEM  |  EWC  |  L2P  | DualPrompt | HiDeLoRA | O-LoRA | TreeLoRA | CoDyRA |
> |-----|:--------:|:-------:|:-----:|:-----:|:-----:|:-----:|:----------:|:--------:|:------:|:--------:|:------:|
> | OP  | 31.16    | 29.73   | 30.12 | 32.19 | 31.96 | 29.38 | 30.76      | 33.73    | 32.94  | 36.14    | **37.46**  |
> | BWT |      -   | 17.03   | 15.2  | 10.74 | 11.62 | 13.57 | 11.34      | 12.36    | 12.89  | 7.36     | **5.11**   |
>
> [2] Wang et al., Trace: A comprehensive benchmark for continual learning in large language models, arXiv:2310.06762.
>
> ---
>
> We sincerely thank the reviewer for the insightful comments and discussion. In response, we have included additional discussions on related works and experiments in the revision (highlighted in blue) to address the concerns. We hope these updates help alleviate the issues raised and kindly ask that you consider revising the score if you find them satisfactory.

---

### Meta-Review · Area_Chair_ZEos · 2026-01-12

**Summary:**

A borderline paper with concern:

- Missing/insufficient comparisons to closest continual-LoRA baselines (TreeLoRA, InfLoRA, CL-LoRA, plus other CLIP/VLM CL methods).
- reviewers were not convinced the rank-based story was theoretically grounded beyond empirical correlations
- concerns about small deltas in headline tables and heavy reliance on CLIP ViT-B/16.
- Forgetting analysis was perceived as incomplete in the original submission.
- The most technically substantive critique: rank minimization is not obviously the right proxy for reduced forgetting without showing reduced directional interference; plus a potential sign typo in the soft-thresholding equations.

**Reviewer Concerns:**

Addressed:

- Missing comparisons (TreeLoRA / CL-LoRA / InfLoRA): added results and updated tables.
- Scalability beyond VLMs (LLMs): added LLaMA-3.2 results on TRACE and positioned them relative to TreeLoRA and others.
- Forgetting analysis: authors added a BWT-based analysis and reported CoDyRA as best retention among the compared set.
- Equation typo (soft-thresholding sign): explicitly acknowledged and corrected; claim implementation was already correct.
- Rank vs interference direction critique: partially addressed via added cosine-similarity/orthogonality-style analysis

Still outstanding:
- The added orthogonality analysis helps, but the “rank minimization -> less forgetting” story still risks being presented too strongly given that interference is fundamentally directional and task-dependent.
- the BWT reporting/interpretation should be defined unambiguously (sign conventions, whether it’s standard BWT or an absolute forgetting proxy)
-

**Reviewer Scores:**

mostly remain unchanged (6, 6, 4, 4)

---

### Decision · Program_Chairs · 2026-01-26

Reject